# Apparent discrepancy of Tibetan ice core $\delta^{18}O$ records may be attributed to misinterpretation of chronology

Shugui Hou[1], Wangbin Zhang[1], Hongxi Pang[1], Shuang-Ye Wu[2], Theo M. Jenk[3,4],

Margit Schwikowski[3,4], Yetang Wang[5]

[1] School of Geographic and Oceanographic Sciences, Nanjing University, Nanjing, 210023, China.

[2] Department of Geology, University of Dayton, Dayton, OH 45469, USA.

[3] Laboratory of Environmental Chemistry, Paul Scherrer Institute, CH-5232 Villigen

PSI, Switzerland.

[4] Oeschger Centre for Climate Change Research, University of Bern, Sidlerstrasse 5, CH-3012 Bern, Switzerland.

[5] College of Geography and Environment, Shandong Normal University, Jinan, China.

Correspondence to: Shugui Hou (shugui@nju.edu.cn)

**Abstract.** Ice cores from the Tibetan Plateau (TP) are widely used for reconstructing

past climatic and environmental conditions that extend beyond the instrumental period.

However, challenges in dating and interpreting ice core records often lead to

inconsistent results. The Guliya ice core drilled from the northwestern TP suggested a

cooling trend during the mid-Holocene based on its decreasing $\delta^{18}O$ values, which is

not observed in other Tibetan ice cores. Here we present a new high-resolution $\delta^{18}O$

record of the Chongce ice cores drilled to bedrock ~30 km away from the Guliya ice

cap. Our record shows a warming trend during the mid-Holocene. Based on our results

as well as previously published ice core data, we suggest that the apparent discrepancy

between the Holocene $\delta^{18}O$ records of the Guliya and the Chongce ice cores may be

attributed to a possible misinterpretation of the Guliya ice core chronology.

**1 Introduction**

Global climate models simulate a warming trend during the Holocene epoch, typically

attributed to retreating ice sheets and rising atmospheric greenhouse gases, while global

cooling was inferred from proxy reconstructions obtained mainly from the analysis of

marine sediment cores (Marcott et al., 2013). The apparent discrepancy is often referred

to as the Holocene temperature conundrum, possibly due to the potentially significant

biases resulted from both the seasonality of the proxy data and the high sensitivities of

current climate models (Liu et al., 2014). Marsicek et al. (2018) recently presented

temperature reconstructions derived from sub-fossil pollen across North America and

Europe. These records show a general long-term warming trend for the Holocene until

~2 kaBP (thousand years before present, present = 1950AD), and records with cooling

trends are largely limited to North Atlantic, implying varied regional climate responses

to global drivers.

Given the significantly positive correlation between air temperature and $\delta^{18}O$ in

precipitation over the central and the northern TP (Yao et al., 1996, 2013), the stable

isotopic records of ice cores recovered from this area were widely used as a temperature

indicator (Tian et al., 2006; An et al., 2016). Among all the published Tibetan ice cores,

the Guliya ice core drilled to bedrock (308.6 m) from the northwestern TP (Fig. S1) is

unique due to the exceptional length of its temporal coverage, estimated to be >500 ka

below the depth of 290 m (i.e., 18.6 m above the ice–bedrock interface), or up to ~760

ka at the ice–bedrock interface based on $^{36}$Cl dead ice in the bottom section (Thompson

et al., 1997). This makes it the oldest non-polar ice core up to now (Thompson et al.,

2017). The Guliya record has been widely used to provide a climate context for

numerous studies (e.g., Fang et al., 1999; Rahaman et al., 2009; Sun et al., 2012; Hou

et al., 2016; Li et al., 2017; Saini et al., 2017; Sanwal et al., 2019). Its stable isotopic

record suggests a cooling mid-Holocene based on its decreasing $\delta^{18}$O values during that

period. However, this cooling mid-Holocene is not found in other Tibetan ice core

records available so far. For instance, the Puruogangri ice core drilled from the central

TP (Fig. S1) shows high $\delta^{18}$O values during the period of ~4.8-4.0 kaBP (Thompson et

al., 2006), and the Dunde ice core drilled from the Qilian mountains (Thompson et al.,

1989; Fig. S1) shows a high stand of $\delta^{18}$O values during the period of ~5-2 kaBP based

on its updated chronology (Thompson et al., 2005). In order to investigate this apparent

discrepancy between the Tibetan $\delta^{18}O$ records, we present a new $\delta^{18}O$ record of the

Chongce ice cores that were recently drilled to bedrock at the Chongce ice cap on the

northwestern TP, ~30 km away from the Guliya ice cap (Hou et al., 2018; Fig. S1).

## 2 The Chongce ice cores and $\delta^{18}O$ measurements

In 2012, we drilled two ice cores to bedrock with the length of 133.8 m (Core 1) and

135.8 m (Core 2, 35º14′ N, 81º7′ E) and a shallow ice core (Core 3) of 58.8 m at an

altitude of 6010 m above sea level (a.s.l.) from the Chongce ice cap (Fig. 1). The

distance between the drilling sites of Core 2 and Core 3 is ~2 m. In 2013, two more ice

cores to bedrock were recovered from the same ice cap with the length of 216.6 m (Core

4, 35°15′ N, 81°5′ E) and 208.6 m (Core 5) at an altitude of 6100 m a.s.l. (Fig. 1). More

details about these ice cores can be found in Hou et al. (2018). For this study,

measurements of stable isotopes were performed on the 135.8 m Core 2 and 58.8 m

Core 3. In a cold room (−20 °C), Core 2 was cut into 1301 samples from the depth of

13.2 m to the bottom with a resolution of ~10 cm/sample. The bottom ~0.2 m above the

ice-bedrock contact consists of a mixture of ice and sediment (Zhang et al., 2018), and

is not analyzed for stable isotopes. The results were combined with the isotopic

measurements of the top 13.2 m of Core 3 from An et al. (2016) to form a single profile

as the two drilling sites are only ~2 m apart. Core 3 has a sampling resolution of 2-3

cm/sample. The samples were measured by a Picarro Wavelength Scanned Cavity Ring-

Down Spectrometer (WS-CRDS, model: L2120-i) at Nanjing University. The stable

isotopic ratio was calculated as:

$$\delta = \left[ \frac{R_{sample}}{R_{reference}} - 1 \right] \times 1000‰$$

where R is the ratio of the composition of the heavier to lighter isotopes in water

($^{18}O/^{16}O$ for $\delta^{18}O$), and the reference is the Vienna Standard Mean Ocean Water (V-

SMOW). Each sample was measured eight times, with the first five measurements

discarded in order to eliminate the effect of memory. The mean value of the last three

measurements was taken as the measurement result. The analytical uncertainty is less

than 0.1‰ for $\delta^{18}O$ (Tang et al. 2015).

**3 Results**

The $\delta^{18}O$ profile by depth of the Chongce ice core is shown in Fig. 2. For comparison,

we also include the depth $\delta^{18}$O profiles of the Guliya (Thompson et al., 1997),

Puruogangri (Thompson et al., 2006) and Dunde (Thompson et al., 1989) ice cores. It

is worth noting that the resolution of the $\delta^{18}$O profiles varies from one ice core to

another. The sampling resolution is ~10 cm/sample for the Chongce Core 2, and 2-3

cm/sample for the Chongce Core 3 (An et al., 2016). The 308.6 m Guliya ice core was

cut into 12628 samples (~2.4 cm/sample) for $\delta^{18}$O measurements (Thompson et al.,

1997). However, the original Guliya data was not available. Instead, the Guliya data

from the NOAA online repository have average resolutions of 10 m, 5 m, 3 m, 1 m and

0.6 m for the depth of 0-100 m, 100-150 m, 150-252 m, 252-308 m and 308-308.6 m

respectively. The 214.7 m Puruogangri ice core was cut into 6303 samples (~3.4

cm/sample) for $\delta^{18}$O measurements (Thompson et al., 2006), but its data from the

NOAA online repository have an average resolution of 1 m for most of the core except

the very top (0-1.05 m) and bottom (214.02-214.7 m) sections. The 139.8 m Dunde ice

core was cut into 3585 samples (~3.9 cm/sample) for $\delta^{18}$O measurements (Thompson

et al., 1989), but its data from the NOAA online repository have an average resolution

of ~1 m for the depth of 0-120 m, and ~0.5 m below 120 m.

**4 Discussion**

The Chongce ice cap has been stable throughout the Holocene, hence provides an ideal

location for retrieving ice cores used to reconstruct past climate. The ice flows from the

Chongce ice cap into the Chongce glacier (Fig. 1). Although the Chongce glacier was

suggested to surge between 1992 and 2014 (Yasuda and Furuya, 2015), it is clear that

the surged area is confined within the Chongce glacier and did not affect the Chongce

ice cap (Fig. 3 of Yasuda and Furuya, 2015). Several other studies have also confirmed

the recent stability of the Chongce ice cap (Lin et al., 2017; Wang et al., 2018; Zhou et

al., 2018). In addition, Wang et al. (2018) found similar mass changes for surge-type

and non-surge-type glaciers over the western Kunlun Mountains, suggesting that the

flow instabilities seem to have little effect on the glacier-wide mass balance. Therefore,

the impact of glacial surge on the stratigraphy of the Chongce ice cap is minimal,

especially in its accumulation zone where our Chongce ice cores were drilled. Over the

longer time scale, Jiao et al. (2000) studied the evolution of glaciers in the west Kunlun

Mountains during the past 32 kaBP. They found that the present terminus of the

Chongce ice cap was very close to its maximum position during the last glacial

maximum (LGM), similar to the Guliya ice cap. This confirms the stability of the

Chongce ice cap since the LGM.

Many studies have shown a significant positive correlation between local temperature

and isotopic composition in precipitation in the northern Tibetan Plateau (e.g., Yao et

al., 1996; Tian et al., 2003). This positive correlation is also observed between local

temperature from instrumental records and isotopic composition in ice cores from

Tibetan Plateau (e.g., Tian et al., 2006; Kang et al., 2007; An et al., 2016). Specifically,

An et al (2016) established a statistically significant correlation between annual $\delta^{18}$O

of Chongce ice core and annual temperature record at Shiquanhe (the nearest climate

station). Although changes in moisture source (e.g., Liu et al., 2015) or large-scale

atmospheric circulation (e.g., Shao et al., 2017) could influence precipitation isotopic

composition in the Tibetan Plateau, such changes often lead to concurrent temperature

change with the same effect on the precipitation isotopes. Therefore, we suggest that

the isotopic variations of Chongce ice core primarily reflect local temperature signals.

There is still uncertainty in the interpretation of the $\delta^{18}$O data of Tibetan ice cores solely

as a temperature proxy across the entire Holocene period. In addition to local

temperature, the precipitation $\delta^{18}O$ could be affected by other factors in longer time-

scales such as changes in the regional circulation patterns, moisture sources and shifts

in seasonal distribution of precipitation (Cheng et al., 2016, Ren et al., 2017). Therefore,

more studies are needed to further examine the validity of using ice core $\delta^{18}O$ as a

temperature proxy on the TP. Nevertheless, simulations by the isotopic general

circulation model (LMDZiso) indicate that a strong positive correlation exists between

the local temperature and precipitation isotope, and it has persisted during the Holocene

(Risi et al., 2010).

Large amplitudes of $\delta^{18}O$ variations are often observed in the Tibetan core cores during

the Holocene, such as ~8 ‰ for the Guliya ice core (Fig. 5) (or ~6‰ based on its

original chronology, Fig. 3), ~6.5 ‰ for the Chongce ice core (Fig. 5), and ~6 ‰ for

the Puruogangri ice core (Fig. 3). This is largely attributed to the elevation dependency

of temperature change observed in many studies, i.e. high altitude regions experience

larger temperature changes than low elevation regions (Beniston et al. 1997; Liu and

Chen, 2000; Mountain Research Initiative EDW Working Group, 2015). In addition,

prominent changes in water vapor sources associated with northward and southward

shifts of the westerly circulation from multi-millennial to orbital timescales (Cheng et

al., 2016) may also contribute to the large amplitude of $\delta^{18}O$ variation in core cores on

TP. However, a sound understanding on the large amplitudes of $\delta^{18}O$ variations requires

comprehensive future work.

A direct comparison of the Tibetan ice core $\delta^{18}O$ records could only be made based on

a common time scale. The chronology of the Chongce, Guliya and Puruogangri ice

cores was established by Hou et al. (2018), Thompson et al. (1997) and Thompson et

al. (2006) respectively. The Dunde ice core was originally dated to be 40 kaBP at the

depth of 5 m above the ice–bedrock interface, and was suggested to be potentially >100

kaBP at the ice–bedrock interface (Thompson et al., 1989). This chronology was

subsequently revised to be within the Holocene (see details in Thompson et al., 2005).

The temporal $\delta^{18}O$ profiles of the Tibetan ice cores are shown in Fig. 3. The $\delta^{18}O$

profiles of the Chongce and Dunde ice cores show an increasing trend from 6-7 kaBP

to ~2.5 kaBP, while the Guliya $\delta^{18}O$ profile shows a decreasing trend from 7 kaBP to

~3 kaBP. The $\delta^{18}O$ profile of the Puruogangri core shows an increasing trend from ~6.5

kaBP to ~4 kaBP,, and remains relatively stable since ~4 kaBP. In addition, the

Grigoriev ice core drilled from the western Tienshan Mountains (see Fig. S1 for

location) also shows a rapid increasing trend of $\delta^{18}O$ since ~8 kaBP (Takeuchi et al.,

2014). Recently, Rao et al. (2019) compiled climatic reconstructions from lake

sediments, loess, sand-dunes and peats in the Xinjiang and surrounding region of

Northwestern China, including northern parts of TP, and brought to the attention the

disagreement between the Guliya ice core and other records. Their reconstructed

records suggest a long-term warming trend during the Holocene. By comparison, it

seems that the $\delta^{18}O$ profile of the Guliya ice core, especially for the period of 6-7 kaBP

to ~3 kaBP, is at odds with this warming trend during the mid-Holocene. It is possible

that this anomaly is not caused by the dramatic difference in local climate conditions,

but linked to the equally anomalous length of Guliya's temporal coverage, which is

over one order of magnitude greater than that of the surrounding ice cores (Hou et al.,

2018). We are aware of studies suggesting a mid-Holocene cooling trend on the TP and

surrounding regions, as argued by Thompson (2019 and references therein). Meanwhile,

other recent studies show a warming trend (e.g., Rao et al., 2019), similar to our results.

Therefore, more research is needed to reach a more definitive conclusion on the Tibetan mid-Holocene climate variations.

Cheng et al. (2012) are one of the first to question the chronology of the Guliya ice core, and argued that it should be shortened by a factor of two (Fig. 4) in order to reconcile the difference in the $\delta^{18}O$ variations between the Guliya ice core and the Kesang stalagmite records (see Fig. S1 for location). However, if compressed linearly by a factor of two, the revised chronology (Guliya-Cheng in Fig. 4) would place the high

Guliya $\delta^{18}O$ values below the depth of 266 m (i.e., 110 kaBP in Fig. 2) in the cold glacial period (North Greenland Ice Core Project members, 2004). This is very unlikely, given the significantly positive relationship between temperature and $\delta^{18}O$ in precipitation over the northwestern TP (Yao et al., 2013; An et al., 2016). We believe the Guliya chronology needs to be further compressed until the high $\delta^{18}O$ values below the depth

of 266 m (i.e., 110 kaBP in Fig. 2) fall within a warm period (Guliya-New in Fig. 4), which is likely to be the mid-Holocene based on the age range of surrounding ice cores (Hou et al., 2018). Since the complete dataset of the Guliya core, as well as its detailed depth-age relationship, is not made available, a detailed comparison between the Guliya

and Chongce ice cores is difficult. Therefore, we attempt to make a direct comparison

between the depth-$\delta^{18}$O profiles of the Guliya and Chongce ice cores. We first divided

the depths of each $\delta^{18}$O data points by the total core length to get the relative depths,

and compared the $\delta^{18}$O profiles of the Guliya and the Chongce ice cores based on their

same relative depth (Fig. 5). The Chongce $\delta^{18}$O profile has much higher sampling

resolution than the publically available Guliya record. In order to account for this

difference, we averaged Chongce $\delta^{18}$O values based on the same relative depth intervals

of the Guliya record as shown in Fig. 2a. After averaging, the Guliya and Chongce $\delta^{18}$O

profiles share much similarity (Fig. 5), and have a highly significant positive correlation

(r=0.57, n=110, p=0.00), whereas their correlation is significantly negative (r=-0.79,

n=16, p=0.00) based on Guliya's original chronology (Fig. 3b). Correlations between

the $\delta^{18}$O profiles of Chongce/Guliya-original and other Tibetan ice cores during their

common period (i.e. 0-6 kaBP) are largely non-significant (Table 1). Although a more

definitive conclusion would require detailed comparison with the original Guliya

dataset (unavailable at the moment) and addition evidence from other Tibetan ice cores,

the highly significant correlation between the Guliya and Chongce $\delta^{18}$O profiles based

on their relative depth suggests the possibility that the Guliya core covers a similar time

span as the Chongce core, which is reasonable given their close proximity (~30 km in

direct distance). Consequently, the apparent discrepancy between the $\delta^{18}O$ records of

the Guliya and other Tibetan ice cores (Fig. 3) may be attributed to a possible

misinterpretation of the Guliya ice core chronology. Although the synchronicity of

glaciation on the TP is beyond the scope of the current work, our new understanding of

the Guliya ice core chronology would cast doubt on using the Guliya record based on

its original chronology as supporting evidence for asynchronous glaciation on the TP

on Milankovitch timescales (Thompson et al., 2005).

Recently, Tian et al. (2019) applied $^{81}$Kr dating, with the updated laser-based detection

method of Atom Trap Trace Analysis (ATTA), to the bottom ice samples collected at

the terminal of the Guliya ice cap. The $^{81}$Kr data yield upper age limits in the range of

15-74 kaBP (90% confidence level). In fact, the exact age is likely to be even younger

than the upper age limits because they lie at the low limit of the ATTA method. The

$^{81}$Kr samples collected at three different sites yielded remarkably consistent results

(Tian et al., 2019), and all the $^{81}$Kr dating results are more than an order of magnitude

younger than the original Chronology of the 1992 Guliya ice core (Thompson et al.,

1997), and roughly in line with the age ranges of the other Tibetan ice cores (Zhang et

al., 2018; Hou et al., 2018).

From September to October of 2015, several new ice cores were recovered from the

Guliya ice cap, including a core to bedrock (309.73 m) and a shallow core (72.40 m)

adjacent to the 1992 Guliya core drilling site, as well as three cores to bedrock (50.72

m, 51.38 m, 50.86 m) from the summit (35º17′ N, 81º29′ E, ~6700 m a.s.l.) of the Guliya

ice cap (Thompson et al., 2018). The Guliya summit 50.80 m ice core (note that the

depth 50.80 m is given in Zhong et al., 2018, which is slightly different from 50.86 m

given in Thompson et al., 2018) was dated to be ~20 kaBP at the depth of 41.10–41.84

m and ~30 kaBP at the depth of 49.51 to 49.90 m by matching the $\delta^{18}O$ values with

those from the 1992 Guliya ice core (Zhong et al., 2018). We made use of the two age

points above, as well as the density profile of the 2015 Guliya summit core (Kutuzov

et al., 2018), to estimate the basal age of the Guliya summit core by applying a two-

parameter flow model (2p model) (Bolzan, 1985), and obtained 76.6 kaBP, 48.6 kaBP

and 42.1 kaBP at the depth of 1 cm w.e., 20 cm w.e. and 40 cm w.e. respectively above

the ice–bedrock contact (Fig. S2). Although these estimates have great uncertainty due

to limited data, the results are still one order of magnitude younger than the original

Chronology of the 1992 Guliya ice core (Thompson et al., 1997) despite the fact that

the two age points (i.e. ~20 kaBP and ~30 kaBP) used by the 2p model are deduced

from the original chronology of the1992 Guliya ice core (Zhong et al., 2018). This casts

further doubt on the original Guliya chronology.

## 5 Conclusions

In this study, we provided a new high-resolution $\delta^{18}O$ record of the Chongce ice cores

drilled from the northwestern TP. Our results show a warming trend for the mid-

Holocene on the TP, which is largely consistent with the Dunde and, to a lesser degree,

Puruogangri ice cores, but much different from the Guliya ice core. It is possible that

the cooling mid-Holocene derived from the Guliya $\delta^{18}O$ record resulted from its

erroneous chronology, rather than the unique boundary conditions on the TP as

previously suggested, such as decreasing summer insolation and weakened Indian

monsoon (Hou et al., 2016; Li et al., 2017). Our study highlighted the urgent need for

more ice core records with reliable chronologies, especially results from the 309.73 m

Guliya ice core drilled in 2015 close to the 1992 Guliya core drilling site (Thompson et

al., 2018) to verify past temperature variation on the TP, which serves as important

baseline information for many other studies, and based on which various scientific

hypotheses such as asynchronous glaciation on the Milankovitch timescales

(Thompson et al., 2005) could be further tested.

Data availability. The $\delta^{18}O$ data of the Chongce ice core are provided in the

Supplement.

Author contributions. SH conceived this study, drilled the Chongce ice cores and

wrote the paper. YW and WZ drilled the Chongce ice cores. WZ performed the $\delta^{18}O$

measurements. All authors contributed to a discussion of the results.

Competing interests. The authors declare that they have no conflict of interest.

**Acknowledgments.** Thanks are due to many scientists, technicians, graduate students

and porters, especially to Yongliang Zhang, Hao Xu and Yaping Liu, for their great

efforts in the high elevations, to Guocai Zhu for providing the ground penetrating radar

results of the Chongce ice cap, and to Lonnie Thompson and Ellen Mosley-Thompson

for sharing the Dunde ice core data. This work was supported by the National Natural

Science Foundation of China (91837102, 41830644, 41711530148, 41330526).

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

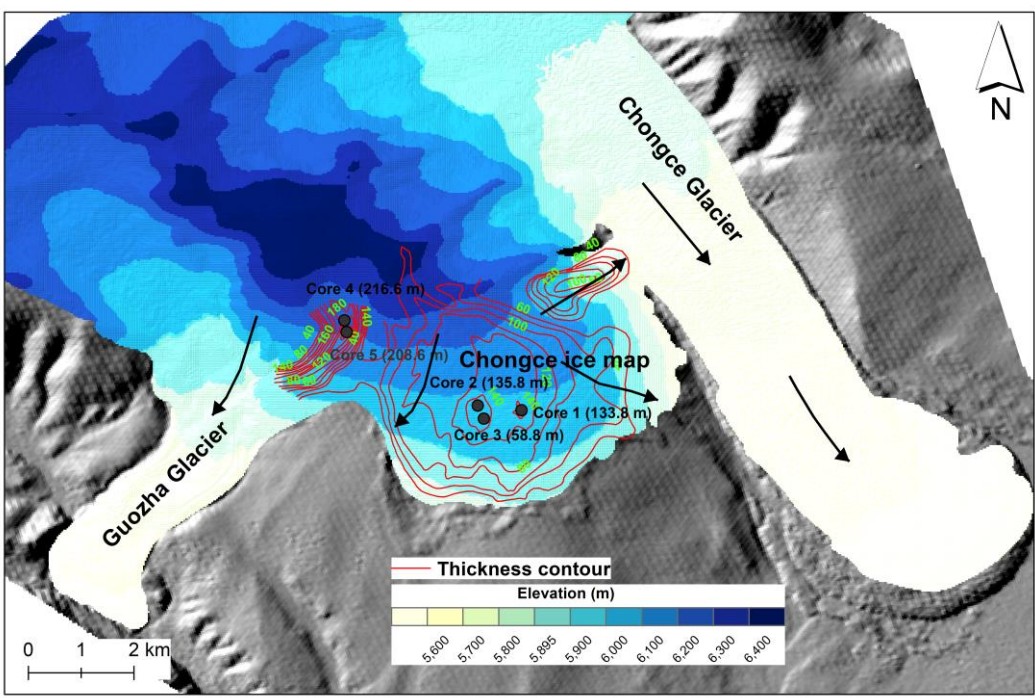


Figure 1. Map showing the topography (red contour lines) and ice thickness (blue color ramp) of the Chongce ice cap with the drilling sites (black dots). The black arrows show the ice flow direction. The effects of the Chongce Glacier surging on the mass balance of the Chongce ice cap is limited, if any (Wang et al., 2018), because the ice flows from

the Chongce ice cap into the Chongce glacier, and the surged area is confined within the Chongce Glacier (Yasuda and Furuya, 2015).

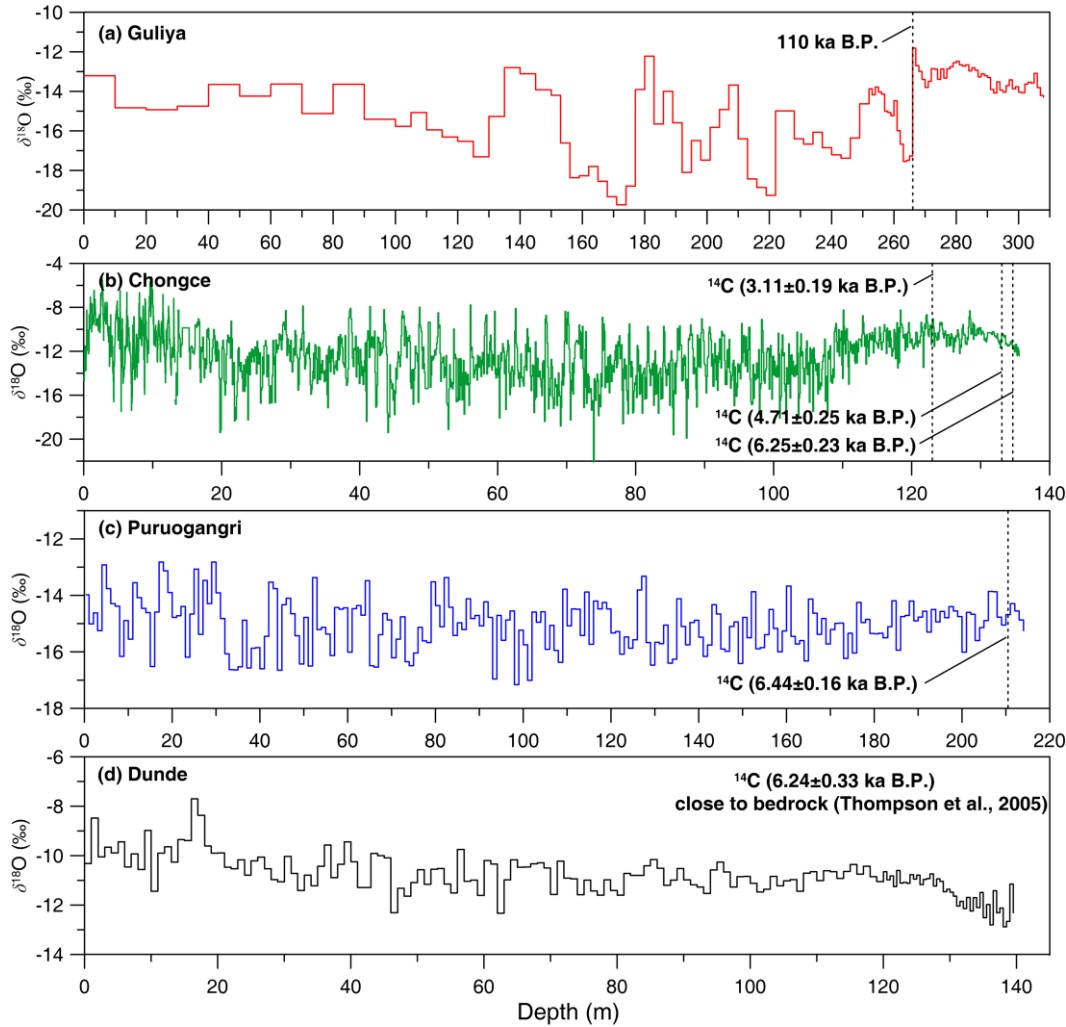

Figure 2. The δ¹⁸O profiles of the ice cores against each respective depth. The age of

110 kaBP at the depth 266 m of the Guliya ice core is from Thompson et al. (1997). The

top 13.2 m of Chongce Core 3 profile (An et al., 2016) is combined with Core 2 to form

a single profile because the distance between their drilling sites is only ~2 m (Fig. 1).

Data of Guliya and Puruogangri were obtained from the NOAA online repository, and

the data of Dunde were extracted from the Fig. 3 in Thompson et al. (1989) using

GetData graph digitizer software.

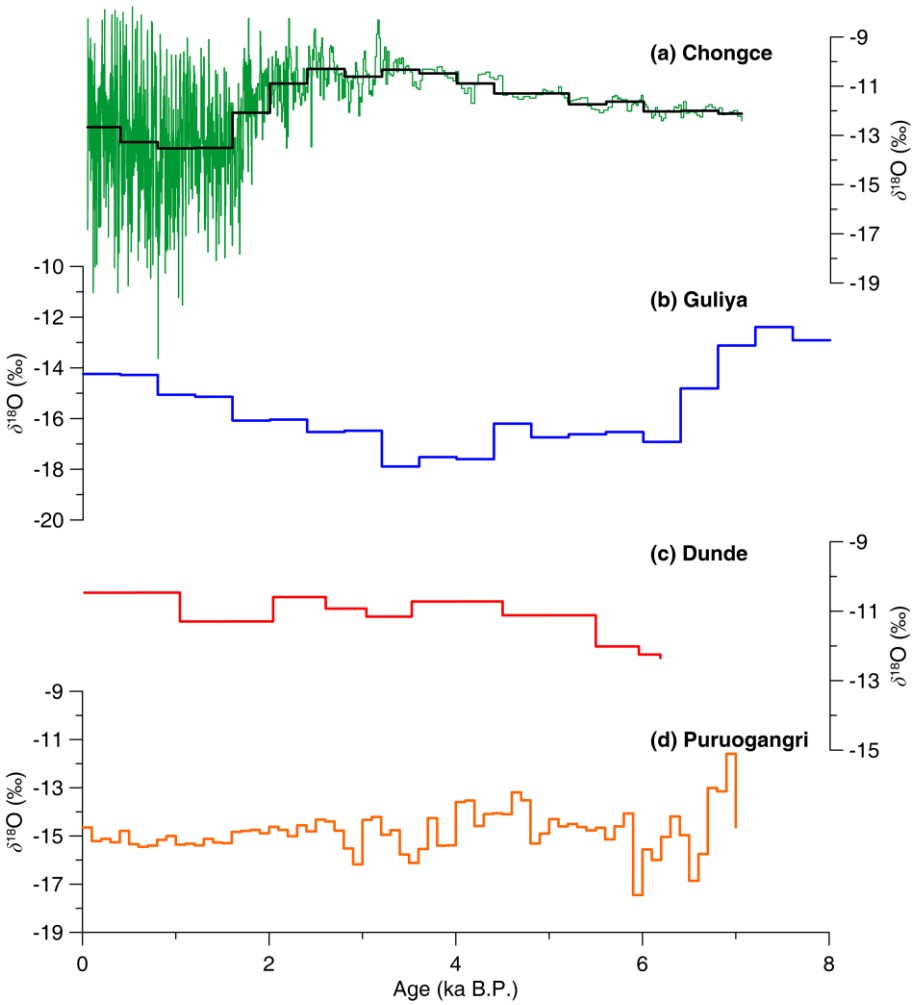

Figure 3. The $\delta^{18}O$ profiles of the Chongce (a), Guliya (b), Dunde (c) and Puruogangri

(d) ice cores by age. We combined the $\delta^{18}O$ profiles of Core 2 and Core 3 into a single

time series. The black line of the Chongce $\delta^{18}O$ profile represents 400-year averages to

match the temporal resolution of the Guliya ice core data that are available from the

NOAA online repository. The 100-year averages of the Puruogangri ice core are also

available from the NOAA online repository, but the multi-centurial averages of the

Dunde ice core were extracted from Figure 3 of Thompson et al. (2005) plotted based

on its updated chronology instead of its original chronology (Thompson et al., 1989).

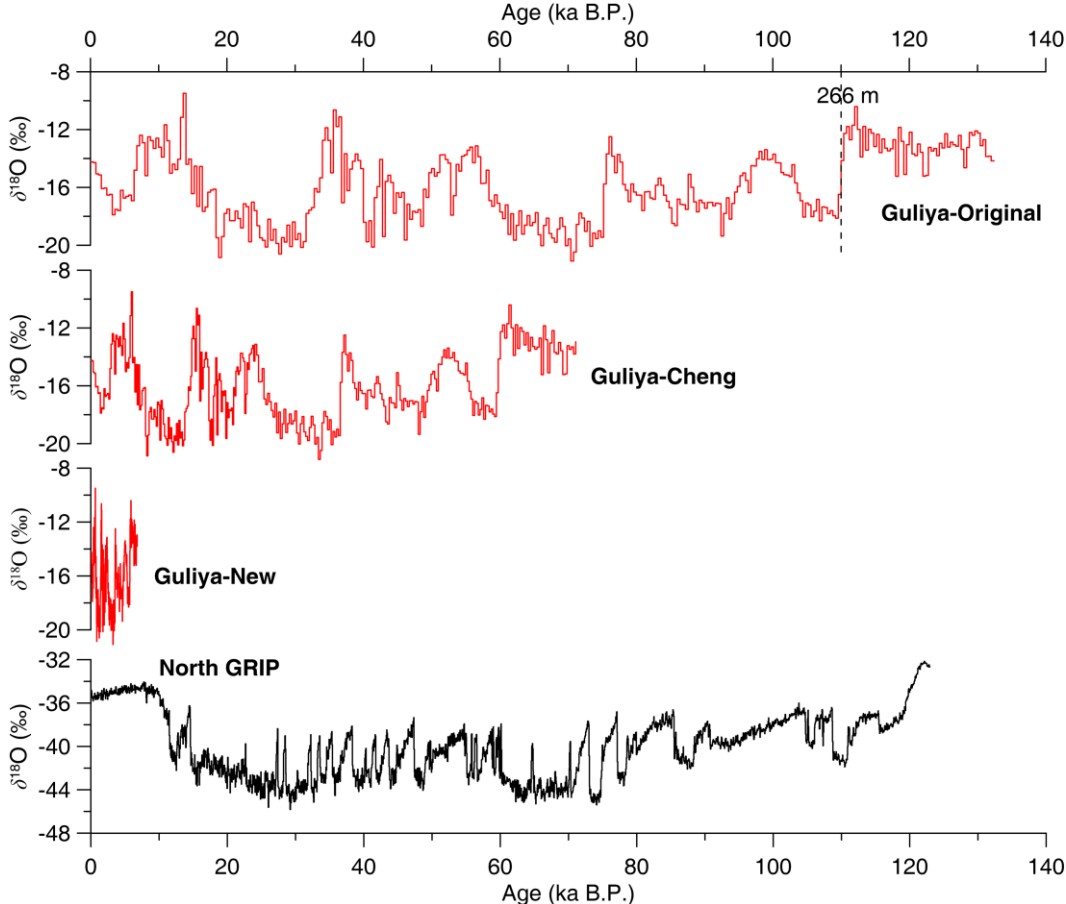

Figure 4. The δ¹⁸O profiles of the Guliya and North GRIP ice cores. The Guliya-Original is plotted on its original chronology (Thompson et al., 1997). The Guliya-Cheng profile is the original Guliya record linearly compressed by a factor of two, as suggested in Cheng et al. (2012). The Guliya-New profile is the original Guliya record further compressed linearly so that the high δ¹⁸O values fall within the warm Holocene.

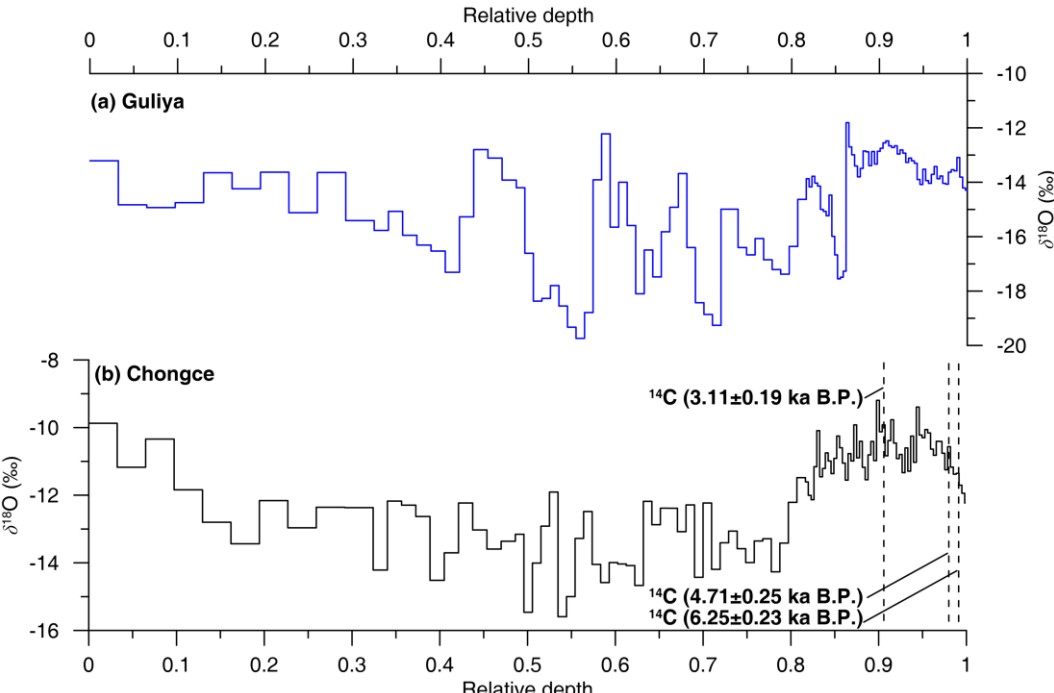

Figure 5. The $\delta^{18}O$ profiles of the Guliya (a) and Chongce (b) ice cores, plotted against

their relative depth. The Chongce profile was averaged to match the temporal resolution

of the published Guliya record as shown in Fig. 2a (Thompson et al., 1997).

Table 1. Correlation coefficients (n=16) between the $\delta^{18}O$ profiles of the Tibetan ice cores.

| | Chongce | Guliya | Puruogangri |
|---|---|---|---|
| Guliya | -0.79 [a] | | |
| Puruogangri | 0.22 | -0.10 | |
| Dunde | 0.11 | 0.24 | -0.11 |

[a] p< 0.001