# Peer review of "Apparent discrepancy of Tibetan ice core $\delta^{18}O$ records may be attributed to misinterpretation of chronology"

_The Cryosphere, 2018_

## Referee Comment (RC1) · Anonymous Referee #1 · 11 Mar 2019

The manuscript entitled "Apparent discrepancy of Tibetan ice core $\delta$18O records may be attributed to misinterpretation of chronology " by Hou et al. presents a new high-resolution $\delta$18O record from the Chongce ice core from the Tibetan Plateau (TP) on the basis of the previously published timescale (Hou et al., 2018). The record covers the middle and late Holocene (the past ∼7 kyr). Although the Chongce ice core is very close to the Guliya ice core (∼30 km away), the Holocene pattern in the Chongce $\delta$18O record is clearly different from the original Guliya $\delta$18O record (Thompson et al., 1997). As such, the authors attributed the observed discrepancy between the Holocene $\delta$18O records of the Guliya and the Chongce ice cores to a misinterpretation of the Guliya ice core chronology. Given the fact that the Guliya record (covering the

past ∼130 kyr based on its original timescale) has been widely used as an important climate reconstruction/benchmark (cited nearly 1000 times), even after its chronology was questioned by Cheng et al. (2012), the new observational data obtained near Guliya and the new insights about Guliya chronology are fascinating and thus deserve to be published. However, I have a few suggestions for improvement pending on which I recommend acceptance of this paper. • The authors imply that they could not get the original dataset of the Guliya and other Tibetan ice core records that were used in several published papers. Please contact the authors of the original papers again to get the original datasets, instead of using digitizer software or other approximate approaches. • The interpretation of Tibetan ice core $\delta$18O data solely as a temperature proxy needs to be further validated. The apparent positive relation observed between ice core $\delta$18O and local temperature from instrumental records cannot be mechanically extrapolated to explain the relation on much longer timescales, for example, the Holocene (e.g., Liu et al., 2015; Shao et al., 2017). This claim is crucial to Tibetan ice core researches, including this paper, and should be more rigorously backed up with empirical data and/or model simulations. • In the past decade, more and more evidences demonstrate that the temporal pattern of the precipitation $\delta$18O changes on orbital-scale, including the Holocene, broadly follows Northern Hemisphere summer insolation (NHSI) inversely in the westerlies (e.g., Bar-Matthews et al., 2003; Cheng et al., 2012a, 2016a; Cai et al., 2017; Mehterian et al. 2017), Indian Monsoon (e.g., Zhang et al., 2011; Cheng et al., 2012b; Cai et al., 2015; Kathayat et al., 2016; Han et al., 2017), East Asian Monsoon (e.g., Cheng et al., 2016b) climatic regimes, as well as within the Tibetan Plateau (e.g., Cai et al., 2010, 2012; Zhang et al., 2011). Cheng et al. (2012) proposed two possibilities: (1) Both the Guliya and Kesang relationships (nearly opposite on orbital-scale) could be valid, with differences related to the different elevations and localities of the sites. (2) Alternately, differences could be reconciled if the low excursions in Guliya $\delta$18O were, instead, correlated to high excursions in CH4 (or higher NHSI). Notably, all aforementioned precipitation $\delta$18O records show a consistent inverse $\delta$18O–NHSI relationship on orbital (possibly millennial) timescale with rather similar amplitudes, in line with the latter possibility. As such, the authors should take the above observations into consideration. In other words, a detail comparison of the Guliya ice core record with the NHSI or a large number of precipitation $\delta$18O records remains one of valid (or better) approaches to establish a more reliable Guliya ice core chronology. Additionally, the new dates from the bottom of the Guliya ice cap indeed show some last glacial ages (Thompson et al., 2018; Zhang et al., 2018; as well as the data in Figure S2), which are consistent with the chronology of the 'Guliya-Cheng' (rather than the 'Guliya-New') reconstructed on the basis of a comparison with other precipitation $\delta$18O records from both Westerlies and Asian Monsoon climatic domains. • Broadly, the amplitude of $\delta$18O variations on orbital (or glacial-interglacial) scale is about ∼8 ‰ in the Westerlies (e.g., Bar-Matthews et al., 2003; Cheng et al., 2012a, 2016a; Mehterian et al., 2017) and Indian Monsoon (e.g., Cai et al., 2010, 2012, 2015; Kathayat et al., 2016) domains, and ∼4 ‰ in the East Asian Monsoon domain (e.g., Cheng et al., 2016b). In addition, the climate during the interglacial time periods, including the Holocene, is fairly stable as inferred by a wide range of proxy records, including various precipitation $\delta$18O records. Provide the 'Gyliya-New' chronology was factual, the prominent multi-millennial changes around the mid-Holocene as characterized by ∼10 ‰ $\delta$18O change (larger than the large regional glacial-interglacial amplitude) would be an unconceivable anomaly (Figure 4), which requires a proper explanation.

References Bar-Matthews et al. 2003. Sea-land oxygen isotopic relationships from planktonic foraminifera and speleothems in the eastern Mediterranean region and their implication for paleorainfall during interglacial intervals. Geochim. Cosmochim. Acta, 67, 3181–3199. Cai et al. 2017. Holocene moisture changes in western China, Central Asia, inferred from stalagmites. Quat. Sci. Rev. 158, 15–28. Cai et al. 2010. Large variations of oxygen isotopes in precipitation over south-central Tibet during Marine Isotope Stage. Geology 38, 243–246. Cai et al. 2012. The Holocene Indian monsoon variability over the southern Tibetan Plateau and its teleconnections. Earth Planet. Sci. Lett. 335-336, 135–144. Cai et al. 2015. Variability of stalagmite-inferred

Indian monsoon precipitation over the past 252,000 y. PNAS 12, 2954–2959. Cheng et al. 2012a. The climatic cyclicity in semiarid-arid central Asia over the past 500,000 years, Geophys. Res. Lett. 39, L01705, https://doi.org/10.1029/2011gl050202. Cheng et al. 2012b. The Global Paleomonsoon as seen through speleothem records from Asia and the Americas. Clim Dyn 39, 1045–1062. Cheng et al. 2016a. Climate variations of Central Asia on orbital to millennial timescales. Sci. Rep. 6:36975 | DOI: 10.1038/srep36975. Cheng et al. 2016b. The Asian monsoon over the past 640,000 years and ice age terminations. Nature 534, 640–646. Han et al. 2017. Climate Change since 7ka BP Revealed by a High-Resolution Stalagmite $\delta$18O and $\delta$13C Record from Benle Cave in Chamdo, Tibet. Acta Geologica Sinica 91, 2545–2556. Kathayat et al., 2016. Indian monsoon variability on millennial-orbital timescales. Sci. Rep. 6:24374 | DOI: 10.1038/srep24374. Liu et al. 2015. Variations in the oxygen isotopic composition of precipitation in the Tianshan Mountains region and their significance for the Westerly circulation. Jour. Geograph. Sci. 25, 801–816. Mehterian et al. 2017. Speleothem records of glacial/interglacial climate from Iran forewarn of future Water Availability in the interior of the Middle East. Quat. Sci. Rev. 164, 187–198. Shao et al. 2017. Driver of the interannual variations of isotope in ice core from the middle of Tibetan Plateau. Atmospheric Research 188, 48–54. Thompson et al. 1997. Tropical climate instability: the last glacial cycle from a Qinghai-Tibetan ice core. Science 276, 1821–1825. Thompson et al. 2018. Ice core records of climate variability on the Third Pole with emphasis on the Guliya ice cap, western Kunlun Mountains. Quat. Sci. Rev. 188, 1–14. Zhang et al. 2011. Holocene monsoon climate documented by oxygen and carbon isotopes from lake sediments and peat bogs in China: a review and synthesis. Quat. Sci. Rev. 30, 1973–1987. Zhong et al. 2018. Clean low-biomass procedures and their application to ancient ice core microorganisms, Front. Microbiol. 9, 1–15.

---

## Referee Comment (RC2) · Lonnie Thompson (Referee) · 11 Mar 2019

Referee Comments on the paper by Hou et al., Apparent discrepancy of Tibetan ice core δ18O records may be attributed to misinterpretation of chronology, for The Cryosphere Discuss., https://doi.org/10.5194/tc-2018-295.

First, it is certainly good to see the recent interest in our work on the Guliya ice core record that was conducted in the 1990s. The community has come a long way since that time when the greatest challenge that Tandong Yao and I faced when drilling in that part of the world was the question of whether or not it would be possible to drill an ice core at those elevations and then keep it frozen during its transit across the Gobi desert. We didn't know at the time how that work would set the stage for all of those who have come along since those early days.

Regarding the time scales on the early Guliya cores, they raised as many questions as they answered and therefore our team returned to Guliya in 2015 where we successfully recovered 5 ice cores, 4 of which were drilled to bedrock. A recently published paper highlights the geophysical work conducted in the field (Kutuzov *et al.*, 2018). A primary goal of the 2015 drilling campaign was to better constrain the time-scale on the Guliya ice cap by taking advantage of additional, newer analytical approaches and applying them to the freshly drilled ice cores. A number of these analyses are focused specifically on dating the ice and are now underway.

Kutuzov, S., L. G. Thompson, I. Lavrentiev, and L. Tian. 2018. Ice thickness measurements of Guliya ice cap, western Kunlun Mountains (Tibetan Plateau), China, *Journal of Glaciology*, 64(248) 977–989, doi: 10.1017/jog.2018.91.

As an invited referee for the paper by Hou *et al.*, I have addressed a number of the specific issues raised in the manuscript but in short the paper lacks sufficient quantitative support for the authors' conclusions. I hope that the following points will help the authors improve their manuscript.

**Specific comments:**

Lines 50-55: *"The Guliya record has been widely used as a benchmark for numerous studies since its publication (e.g., Fang et al., 1999; Rahaman et al., 2009; Sun et al., 2012; Hou et al., 2016; Li et al., 2017; Saini et al., 2017; Sanwal et al., 2019). Its stable isotopic record suggests a cooling mid-Holocene based on its decreasing $\delta^{18}O$ values during that period. However, this cooling mid-Holocene is not found in other Tibetan ice core records available so far."*

The first sentence will be addressed below. The third sentence is misleading. The mid-Holocene cooling is very noticeable in Tibetan climate records that are not from ice cores. For example, the regional vegetation and climate changes during the Holocene have been reconstructed from a high-resolution pollen record preserved in a peat sequence from the Altai Mountains of Xinjiang, China (Zhang *et al.*, 2018, *Quaternary Science Reviews*, 201, 111-123). These vegetation phases indicate that the regional climate changed from a cold and dry early Holocene to a warmer and wetter early-mid Holocene followed by a cold and dry mid-Holocene, which transitioned to a cool and wet late Holocene with warm and dry conditions characterizing the last millennium. Below is a figure comparing the data in Figure 6 of the Zhang *et al.* paper (left) with Figure 3 (right) from the Hou *et al.* paper. Note that the Guliya $\delta^{18}O$ record (blue) is more similar to the

mean annual temperature (Figure 6, panel f, red star) than the Chongce $\delta^{18}$O record. It is also important to note that the Guliya ice core was not used to help establish the chronology of the pollen record.

[Figure]

The figure is a composite of Figure 6 (Zhang *et al.*, 2018) and Figure 3 (Hou *et al*., unpublished).

The records above, along with other examples given below, dispute Lines 136-140 *("This warming trend during the mid-Holocene is similar to recent paleoclimatic reconstructions in other parts of the world (Samartin et al., 2017; Marsicek et al., 2018). By comparison, it seems that the $\delta^{18}$O profile of the Guliya ice core, especially for the period of 6-7 kaBP to ~3 kaBP, is at odds with this warming trend during the mid-Holocene.")*. Here the authors are picking records from regions thousands of miles away in much different climate regimes to confirm the Chongce $\delta^{18}$O record (and time scale). The Samartin *et al*. records are from the Mediterranean while the Marsicek *et al*. records are from Europe and North America. Hou *et al*. (Lines 35-40) state that *"Marsicek et al. (2018) recently presented temperature reconstructions derived from sub-fossil pollen across North America and Europe. These records show a general long-term warming trend for the Holocene until ~2 kaBP (thousand years before present),and records with cooling trends are largely limited to North Atlantic, implying varied regional climate responses to global drivers")*. There are several publications that link North Atlantic climate to the climates of Central Asia and China. Although most of them discuss the linkages between precipitation and westerlies influenced by North Atlantic atmospheric and oceanic processes, papers such as Feng and Hu (2008, *Geophysical Research Letters* 35 doi: 10.1029/2007GL032484) present an argument that North Atlantic SST anomalies strongly affect the TP surface temperature and heat sources, at least in the last century.

There are other records that call into question their conclusions regarding Holocene climate variability as inferred from the Chongce cores. For example, Zhang and Feng (*Earth-Science Reviews*, 2018, 185, 847-869) presented a compilation of pollen records from the Altai Mountains and surrounding regions that show a mid-Holocene cooling trend. Below see their Figure 37 (note panel d) from their synthesis of regional pollen records.

[Figure]

This is Figure 37 from Zhang and Feng, 2018 which was cited above.

Another example that does not support the conclusions drawn from the Chongce ice core is an alkenone-based 21 ka paleotemperature record from Lake Balikun (43.60-43.73ºN, 92.74-92.84ºE, 1570 masl). As shown in the figure below (see panel d), this lake record shows that in this region the peak summer temperature occurred at 8 ka and was followed by general cooling throughout the Holocene.

[Figure]

This is Figure 8 is from Zhao *et al*. 2017 (Contrasting early Holocene temperature variations between monsoonal East Asia and westerly dominated Central Asia. *Quaternary Science Reviews* 178, 14-23).

Warmer conditions for the Early Holocene and cooler temperatures in the mid-Holocene are inferred by additional eastern TP records (see papers cited below). Many of these records are consistent with the Northern Hemisphere summer insolation curve (see panel a in the figure above from Zhang and Feng, 2018).

Shen, J., Liu, X., Wang, S., Ryo, M., 2005. Palaeoclimatic changes in the Qinghai Lake area during the last 18,000 years. *Quaternary International* 136, 131–140.

Yu, X., Zhou, W., Franzen, L.G., Xian, F., Cheng, P., Jull, A.J.T., 2006. High-resolution peat records for Holocene monsoon history in the eastern Tibetan Plateau. *Science in China (Series D)* 49, 615–621.

Herzschuh, U., Kramer, A., Mischke, S., Zhang, C., 2009. Quantitative climate and vegetation trends since the late glacial on the northeastern Tibetan Plateau deduced from Koucha Lake pollen spectra. *Quaternary Research* 71, 162–171.

Zhang, C., Mischke, S., 2009. A Late Glacial and Holocene lake record from the Nianbaoyeze Mountains and inferences of lake, glacier and climate evolution on the eastern Tibetan Plateau. *Quaternary Science Reviews* 28, 1970–1983.

Kramer, A., Herzschuh, U., Mischke, S., Zhang, C., 2010. Holocene tree line shifts and monsoon variability in the Hengduan Mountains (southeastern Tibetan Plateau), implications from palynological investigations. *Palaeogeography, Palaeoclimatology, Palaeoecology* 286, 23–41.

On *Lines 50 - 53* the authors falsely state that "*The Guliya record has been widely used as a benchmark for numerous studies since its publication (e.g., Fang et al., 1999; Rahaman et al., 2009; Sun et al., 2012; Hou et al., 2016; Li et al., 2017; Saini et al., 2017; Sanwal et al., 2019).*

*Returning to Lines 50-54,* The definition of "benchmark" is a point of reference from which measurements may be made. In none of the references cited above are the time series constructed to match that of Guliya. Those chronologies were independently developed. Therefore the suggestion that the Guliya record misled the development of the climate records in these or any other papers is false. This sentence should be rephrased as "The Guliya record has been compared with climate records from numerous studies….."). The records in these and other references were broadly compared to the Guliya record. If the climate records from these independently dated records match the Guliya record then it is not because they *were matched to* Guliya in order to develop their chronologies, it is because their independent chronologies were coherent with the Guliya chronologies. Also, if the Holocene temperature records presented in these publications are similar to Guliya's Holocene $\delta^{18}O$ (temperature) time series, which contradicts the Chongce $\delta^{18}O$ (temperature) record, it raises a serious challenge to the validity of the interpretation of the Chongce records, which the authors should address.

Hou *et al.* make statements that are inconsistent with existing evidence. For example they state (Line 179-181): "*This would also cast doubt on the notion of asynchronous glaciation on the TP on Milankovitch timescales (Thompson et al., 2005), which is developed based on the original chronology of the Guliya ice core.*"
Guliya is not the solitary piece of evidence supporting asynchronous glaciation on the Tibetan Plateau. There are a number of exposure dates that also point to asynchronous glaciation. Owen *et al.* (2008, Quaternary glaciation of the Himalayan-Tibetan orogeny in *J. Quaternary Science* 23, 513-531) state in their abstract "Glaciers throughout monsoon-influenced Tibet, the Himalaya and the Transhimalaya are likely synchronous both with climate change resulting from

oscillations in the South Asian monsoon and with Northern Hemisphere cooling cycles. In contrast, glaciers in Pamir in the far western regions of the Himalayan–Tibet orogen advanced asynchronously relative to the other regions that are monsoon-influenced regions and appear to be mainly in phase with the Northern Hemisphere cooling cycles."

Lines 182-184: *Recently, Ritterbusch et al. (2018) applied 81Kr dating, with the updated laser-based detection method of Atom Trap Trace Analysis (ATTA), to the bottom ice samples collected at the terminal of the Guliya ice cap. The resulting 81Kr ages are <50 kaBP.*  [81]Kr ages on the margin of the Guliya ice cap tell us nothing about the age of the bottom ice of the 308m ice core at the Plateau "Site 2" drill site (where the 1992 core was drilled). Ice samples collected in 2015 for [81]Kr analyses were collected down the flowline and in close proximity to our 1992 Site 1 drill (see locations in Figure 1 of Thompson *et al.*, 1995, *Annals of Glaciology*). In 1992 the first Guliya core "Site 1" was drilled to 92.2 meters, at which point we terminated drilling because we found an unconformity in the ice layers 83 meters below the surface (see discussion on page 176 in the aforementioned Thompson *et al.* 1995 paper). Thus, there is no reason to believe there is a time stratigraphic linkage between the bottom ice along the margin (near the camp, see aforementioned map) and the ice at the bottom of our deep core drilled on the Plateau at Site 2 (see map).

Minor points

Some statements are erroneous or misleading and need to be checked and verified. For example, on *Lines 128-130 they state: "However, this high $\delta^{18}O$ value is not observed around the depth of ~211 m in the Puruogangri depth $\delta18O$ profile (Fig. 2). Indeed, all $\delta18O$ values in the depth profile of the Puruogangri core are well below -12‰. Therefore, the high $\delta^{18}O$ value around ~7 kaBP of the Puruogangri core (Fig. 3) needs further verification."* Those values exist in the raw data around 211 meters (the raw data below are ~ 6.9-7.0 ka), and this high $\delta^{18}O$ value is a function of the time averaging (100 yr averages), whereas the authors are basing their observations on one meter averages, which incorporate ~30 data points).

| Depth (m) | $\delta^{18}O$ (‰) |
|-----------|---------------------|
| 210.960   | -11.35              |
| 210.990   | -11.30              |
| 211.025   | -12.12              |

Finally, the authors' failed to mention that evidence exists suggesting that Chongce may be a surging glacier. In 1991 Chinese scientists published a Quaternary Glacial Distribution Map of the Tibetan Plateau. According to this map, the terminal moraines around the Guliya ice cap are very close to their maximum position during the last two glaciations. However, this is not the case for the Chongce ice cap which shows the greatest variations in ice extent of any of the ice caps in this region. In addition, the Chongce glacier, which flows from the Chongce ice cap, surged between 1992 and 2014 while the Guliya ice cap remained static (Yasuda and Furuya, 2015; Fig. 3). Therefore, it might be inaccurate to assume that the timescale developed for the Chongce cores should reflect that of Guliya. In light of the geophysical considerations discussed

above it is premature to conclude that the Chongce results invalidate the much longer Guliya timescale.

Yasuda, T. and Furuya, M. 2015. Dynamics of surge-type glaciers in West Kunlun Shan, Northwestern Tibet. *Journal of Geophysical Research - Earth Surface*, https://doi.org/10.1002/2015JF003511.

Note to readers of this review:
When asked by Editor Carlos Martin to serve as a referee for this paper, I inquired whether this would constitute a conflict of interest as our Guliya record is a major subject of the paper. I was told "My view is that there is no conflict of interest". Therefore, I opted to serve as a referee.

---

## Author Comment (AC1) · 15 Apr 2019

The manuscript entitled "Apparent discrepancy of Tibetan ice core $\delta^{18}O$ records may be attributed to misinterpretation of chronology" by Hou et al. presents a new high resolution $\delta^{18}O$ record from the Chongce ice core from the Tibetan Plateau (TP) on the basis of the previously published timescale (Hou et al., 2018). The record covers the middle and late Holocene (the past ~7 kyr). Although the Chongce ice core is very close to the Guliya ice core (~30 km away), the Holocene pattern in the Chongce $\delta^{18}O$ record is clearly different from the original Guliya $\delta^{18}O$ record (Thompson et al., 1997). As such, the authors attributed the observed discrepancy between the Holocene $\delta^{18}O$ records of the Guliya and the Chongce ice cores to a misinterpretation of the Guliya ice core chronology. Given the fact that the Guliya record (covering the past ~130 kyr based on its original timescale) has been widely used as an important climate reconstruction/benchmark (cited nearly 1000 times), even after its chronology was questioned by Cheng et al. (2012), the new observational data obtained near Guliya and the new insights about Guliya chronology are fascinating and thus deserve to be published. However, I have a few suggestions for improvement pending on which I recommend acceptance of this paper.

(1) The authors imply that they could not get the original dataset of the Guliya and other Tibetan ice core records that were used in several published papers. Please contact the authors of the original papers again to get the original datasets, instead of using digitizer software or other approximate approaches.

Response:

I sent an email on 3 April to the corresponding author of the original papers regarding the possibility of sharing the original datasets of the Guliya and other Tibetan ice core records, and got responses from Prof. Lonnie Thompson on 13 April, and Prof. Ellen Mosley-Thompson on 15 April. They said they would provide a web link for downloading the Dunde ice core $\delta^{18}O$ datasets. We are very grateful for their willingness to share the datasets, and will update the figures accordingly with the datasets. It is worth pointing out that, even without the original datasets, the general patterns of the Guliya and Dunde $\delta^{18}O$ profiles are sufficiently preserved in the summary data to support our conclusions.

(2) The interpretation of Tibetan ice core $\delta^{18}O$ data solely as a temperature proxy needs to be further validated. The apparent positive relation observed between ice core $\delta^{18}O$ and local temperature from instrumental records cannot be mechanically extrapolated to explain the relation on much longer timescales, for example, the Holocene (e.g., Liu et al., 2015; Shao et al., 2017). This claim is crucial to Tibetan ice core researches, including this paper, and should be more rigorously backed up with empirical data and/or model simulations.

Response:

Many studies have shown a significant positive correlation between local temperature and isotopic composition in precipitation in the northern Tibetan Plateau (e.g., Yao et al., 1996; Tian et al., 2003). This positive correlation is also observed between local temperature from instrumental records and isotopic composition in ice cores from Tibetan Plateau (e.g., Tian et al., 2006; Kang et al., 2007; An et al., 2016). Specifically, An et al (2016) established a statistically significant correlation between annual (not seasonal) $\delta^{18}O$ of Chongce ice core and annual temperature record at Shiquanhe (the nearest climate station). In addition, simulations by the LMDZ4 general circulation model indicate that this positive correlation between local temperature and precipitation isotope has persisted during the Holocene (Risi et al., 2010).

Although changes in moisture source (as indicated by Liu et al., 2015) or large-scale atmospheric circulation (as indicated by Shao et al., 2017) could influence precipitation isotopic composition in the Tibetan Plateau, such changes often lead to concurrent temperature change with the same effect on the precipitation isotopes. Therefore, we believe that the isotopic variability of Chongce ice core primarily reflects local temperature signals.

(3) In the past decade, more and more evidences demonstrate that the temporal pattern of the precipitation $\delta^{18}O$ changes on orbital-scale, including the Holocene, broadly follows Northern Hemisphere summer insolation (NHSI) inversely in the westerlies (e.g., Bar-Matthews et al., 2003; Cheng et al., 2012a, 2016a; Cai et al., 2017; Mehterian et al. 2017), Indian Monsoon (e.g., Zhang et al., 2011; Cheng et al., 2012b; Cai et al., 2015; Kathayat et al., 2016; Han et al., 2017), East Asian Monsoon (e.g., Cheng et al., 2016b) climatic regimes, as well as within the Tibetan Plateau (e.g., Cai et al., 2010, 2012; Zhang et al., 2011). Cheng et al. (2012) proposed two possibilities: (1) Both the Guliya and Kesang relationships (nearly opposite on orbital-scale) could be valid, with differences related to the different elevations and localities of the sites. (2) Alternately, differences could be reconciled if the low excursions in Guliya $\delta^{18}O$ were, instead, correlated to high excursions in $CH_4$ (or higher NHSI). Notably, all aforementioned precipitation $\delta^{18}O$ records show a consistent inverse $\delta^{18}O$–NHSI relationship on orbital (possibly millennial) timescale with rather similar amplitudes, in line with the latter possibility. As such, the authors should take the above observations into consideration. In other words, a detail comparison of the Guliya ice core record with the NHSI or a large number of precipitation $\delta^{18}O$ records remains one of valid (or better) approaches to establish a more reliable Guliya ice core chronology. Additionally, the new dates from the bottom of the Guliya ice cap indeed show some last glacial ages (Thompson et al., 2018; Zhang et al., 2018; as well as the data in Figure S2), which are consistent with the chronology of the 'Guliya-Cheng' (rather than

the 'Guliya-New') reconstructed on the basis of a comparison with other precipitation $\delta^{18}O$ records from both Westerlies and Asian Monsoon climatic domains.

Response:

We think that the Guliya-New chronology is more reasonable than Guliya-Cheng for several reasons. (1) The Guliya-Cheng chronology would put the high stands of $\delta^{18}O$ values of the Guliya profile from the depth 266 m to the ice core bottom (Fig. 4 in our manuscript) in the cold glacial period. This is very unlikely, given the significantly positive relationship between temperature and $\delta^{18}O$ in precipitation over the northwestern TP (see the response above). (2) The ages established in Zhang et al (2018) and Ritterbusch et al. (2018) only serve to provide upper constraints, and the actual bottom age of the ice cores is likely to be younger. Thompson et al. (2018) did not provide any new estimates of the bottom age of the Guliya ice cores (both 1992 and 2015 cores), as they wrote that "Future analyses will include [14]C on organic material trapped in the ice, and [36]Cl, beryllium-10 ([10]Be), $\delta^{18}O$ of air in bubbles trapped in the ice, and argon isotopic ratios ([40]Ar/[38]Ar) on deep sections of 2015PC2 to determine more precisely the age of the ice cap." (3) The data in Fig. S2 in our manuscript is based on Zhong et al. (2018), who established the chronology of the 2015 Guliya summit ice core by matching its $\delta^{18}O$ values with those from the 1992 Guliya ice core (Thompson et al., 1997). There is still much inconsistency between the age ranges of the 2015 Guliya summit ice core and the1992 Guliya ice core despite the fact that the two age points of the 2015 Guliya summit ice core are deduced from the original chronology of the1992 Guliya ice core. This casts further doubt on the original 1992 Guliya chronology. Consequently, the chronology of the 2015 Guliya summit ice core might also suffer from this questionable original 1992 Guliya chronology. (4) Hou et al. (2018) provided convincing evidence that the bottom age of the Chongce ice cores is likely within the Holocene, consistent to the other Tibetan ice cores except the Guliya ice core. Given the similarity between the Guliya and Chongce depth $\delta^{18}O$ profiles (Fig. 5 in our manuscript), it is reasonable to suggest that the Guliya core covers a similar time span as the Chongce core, though a more detailed comparison (Fig. 4 in our manuscript) would be necessary when more evidence and the original datasets of the Tibetan ice cores become available in order to confirm the Guliya-New chronology.

Consistent with all other precipitation $\delta^{18}O$ records in the westerlies regime, the Chongce ice core $\delta^{18}O$ record also shows an inverse $\delta^{18}O$-NHSI relationship at the precession time scales. There are two possible explanations for this inverse $\delta^{18}O$–NHSI relationship. First, some studies suggest this inverse relationship is caused by the possible incursions of the Asian summer monsoon moisture (with low $\delta^{18}O$) into central Asia during the high NHSI summers. For example, the speleothem $\delta^{18}O$ record from Kesang Cave in

northwestern China was much depleted at times of high NHSI (Cheng et al., 2012, 2016), a feature closely resembling speleothem records in Asian summer monsoon regime. The second explanation suggests that one would expect an inverse $\delta^{18}$O–NHSI relationship if winter precipitation (with low $\delta^{18}$O) in the westerlies region increased during the low Northern Hemisphere winter insolation (NHWI, which has a reverse phase with NHSI) (Tzedakis, 2007; Kutzbach et al., 2014). At present, there is no consensus on what caused the inverse $\delta^{18}$O-NHSI relationship, and additional studies are needed for unravelling the underlying mechanisms. Here, we compared the Chongce isotopic record with other records of precipitation $\delta^{18}$O in the westerlies regime, including speleothem $\delta^{18}$O records from the Kesang Cave in the northwestern China (Cheng et al., 2012), the Ton Cave in Uzbekistan (Cheng et al., 2016), the Kinderlinskaya Cave in the southern Ural Mountains (Baker et al., 2017), and the Soreq Cave from Central Israel (Bar-Matthews et al., 2003), and a record of the oxygen isotope composition of permafrost ice wedges from the Lena River Delta in the Siberian Arctic (Meyer et al., 2015) (Fig. 1). All of these records show a consistent rising trend during the middle to late Holocene, in contrast with decreasing trend observed in the isotopic record of the Guliya ice core during this period.

[Figure]

**Fig. 1:** Comparison of oxygen isotopic records during the Holocene from the Chongce ice core (a), the Kesang Cave (Cheng et al., 2012) (b), the Ton Cave in Uzbekistan (Cheng et al., 2016) (c), the Kinderlinskaya Cave in the southern Ural Mountains (Baker et al., 2017) (d), the Soreq Cave from Central Israel (e) and permafrost ice wedges from the Lena River Delta in the Siberian Arctic (Meyer et al., 2015) (f).

(4) Broadly, the amplitude of $\delta^{18}O$ variations on orbital (or glacial-interglacial) scale is about ~8 ‰ in the Westerlies (e.g., Bar-Matthews et al., 2003; Cheng et al., 2012a, 2016a; Mehterian et al., 2017) and Indian Monsoon (e.g., Cai et al., 2010, 2012, 2015; Kathayat et al., 2016) domains, and ~4 ‰ in the East Asian Monsoon domain (e.g., Cheng et al., 2016b). In addition, the climate during the interglacial time periods, including the Holocene, is fairly stable as inferred by a wide range of proxy records, including various precipitation $\delta^{18}O$ records. Provide the 'Guliya-New' chronology was factual, the prominent multi-millennial changes around the mid-Holocene as characterized by ~10 ‰ $\delta^{18}O$ change (larger than

the large regional glacial-interglacial amplitude) would be an unconceivable anomaly (Figure 4), which requires a proper explanation.

Response: Large amplitudes of $\delta^{18}$O variations are often observed in the Tibetan core cores during the Holocene, such as ~8 ‰ for the Guliya ice core (Fig. 2 in the manuscript) (or ~6‰ based on its original chronology, Fig. 3 in the manuscript), ~6.5 ‰ for the Chongce ice core (Fig. 5 in the manuscript), and ~6 ‰ for the Puruogangri ice core (Fig. 3 in the manuscript). This is largely attributed to the elevation dependency of temperature change observed in many studies, i.e. high altitude regions experience larger temperature changes than low elevation regions (Beniston et al. 1997; Liu and Chen, 2000; Mountain Research Initiative EDW Working Group, 2015). In addition, prominent changes in water vapor sources associated with northward and southward shifts of the westerly circulation on longer timescale (e.g., from multi-millennial to orbital timescales) may also contribute to the large amplitude of $\delta^{18}$O variation in core cores on the Tibetan Plateau.

References cited:

An, W., Hou, S., Zhang, W., Wu, S., Xu, H., Pang, H., Wang, Y., and Liu, Y.: Possible recent warming hiatus on the northwestern Tibetan Plateau derived from ice core records, Sci. Rep., 6:32813, doi:10.1038/srep32813, 2016.

Baker, J. L., Lachniet, M. S., Chervyatsova, O., Asmerom, Y., and Polyak, V. J.: Holocene warming in western continental Eurasia driven by glacial retreat and greenhouse forcing, Nat. Geosci., doi:10.1038/NGEO2953, 2017.

Bar-Matthews, M., Ayalon, A., Gilmour, M., Matthews, A., and Hawkesworth, C. J.: Sea-land oxygen isotopic relationships from planktonic foraminifera and speleothems in the eastern Mediterranean region and their implication for paleorainfall during interglacial intervals, Geochim. Cosmochim. Acta, 67, 3181–3199, 2003.Beniston, M., Diaz, H. F., Bradley, R. S.: Climatic change at high elevation sites: an overview, Clim. Change, 36, 233–251, 1997.

Cheng, H., Zhang, P., Spötl, C., Edwards, R. L., Cai, Y., Zhang, D., Sang, W., Tan, M., and An, Z.: The climatic cyclicity in semiarid-arid central Asian over the past 500,000 years, Geophy. Res. Lett., 39, L01705, doi:10.1029/2011GL050202, 2012.

Cheng, H., Spötl, C., Breitenbach, S. F. M., Sinha, A., Wassenburg, J. A., Jochum, K. P., Scholz, D., Li, X., Yi, L., Peng, Y., Lv, Y., Zhang, P., Votintseva, A., Loginov, V., Ning, Y., Kathayat, G., and Edwards, R. L.: Climate variations of central Asia on orbital to millennial timescales, Sci. Rep., 6:36975, doi:10.1038/srep36975, 2016.

Hou, S., Jenk, T., Zhang, W., Wang, C., Wu, S., Wang, Y., Pang, H., and Schwikowski, M.: Age ranges of the Tibetan ice cores with emphasis on the Chongce ice cores, western Kunlun Mountains, The Cryosphere 12, 2341–2348, doi: 10.5194/tc-12-2341-2018, 2018.

Kang, S., Zhang, Y., Qin, D., Ren, J., Zhang, Q., Grigholm, B., and Mayewski, P.: Recent temperature increase recorded in an ice core in the source region of Yangtze River. Chin. Sci. Bull., 52, 825–831, doi:10.1007/s11434-007-0140-1, 2007.

Kutzbach, J. E., Chen, G., Cheng, H., Edwards, R. L., Liu, Z.: Potential role of winter rainfall in explaining increased moisture in the Mediterranean and Middle East during periods of maximum orbitally-forced insolation seasonality, Clim. Dyna., 42, 1079-1095, 2014.

Liu, X. and Chen, B.: Climatic warming in the Tibetan Plateau during recent decades, Int. J. Climol. 20, 1729–1742, 2000.

Liu, X., Rao, Z., Zhang, X., Huang, W., Chen, J., and Chen, F.: Variations in the oxygen isotopic composition of precipitation in the Tianshan Mountains region and their significance for the Westerly circulation. J. Geography Sci. 25, 801–816, 2015.

Maussion, F., Scherer, D., Mölg, T., Collier, E., Curio, J., and Finkelnburg, R.: Precipitation seasonality and variability over the Tibetan Plateau as resolved by the high Asia reanalysis, J. Clim., 27, 1910-1927, 2014.

Meyer, H., Opel, T., Laepple, T., Dereviagin, A., Hoffmann, K., and Werner, M.: Long-term winter warming trend in the Siberian Arctic during the mid-to late Holocene, Nat. Geosci., doi:10.1038/NGEO2349, 2015.

Mountain Research Initiative EDW Working Group.: Elevation-dependent warming in mountain regions of the world, Nat. Clim. Change, 5, 424-430, 2015.

Risi, C., Bony, S., Vimeux, F., and Jouzel, J.: Water stable isotopes in the LMDZ4 General Circulation Model: Model evaluation for present day and past climates and applications to climatic interpretation of tropical isotopic records, J. Geophys. Res., 115, D12118, doi:10.1029/2009jd013255, 2010.

Ritterbusch, F., Chu, Y., Dong, X., Gu, J., Hu, S., Jiang, W., Landais, A., Lipenkov, V., Lu, Z., Shi, G., Stenni, B., Taldice team, Tian, L., Tong, A., Yang, G., and Zha, L.: Revealing old ice with [81]Kr, Geophysical Research Abstracts. 20, EGU2018-2350-5, 2018.

Shao et al.: Driver of the interannual variations of isotope in ice core from the middle of Tibetan Plateau, Atmos. Res., 188, 48–54, 2017.

Thompson, L. G., Yao, T., Davis, M. E., Henderson, K. A., Mosley-Thompson, E., Lin, P.-N., Beer, J., Synal, H.-A., Cole-Dai, J., and Bolzan, J. F.: Tropical climate instability: the last glacial cycle

from a Qinghai-Tibetan ice core, Science, 276, 1821-1825, doi: 10.1126/science.276.5320.1821, 1997.

Thompson, L., Yao, T., Davis, M., Mosley-Thompson, E., Wu, G., Porter, S., Xu, B., Lin, P., Wang, N., Beaudon, E., Duan, K., Sierra-Hernández, M., and Kenny, D.: Ice core records of climate variability on the Third Pole with emphasis on the Guliya ice cap, western Kunlun Mountains, Quat. Sci. Rev., 188, 1–14, doi: 10.1016/j.quascirev.2018.03.003, 2018.

Tian, L., Yao, T., Schuster, P. F., White, J. W. C., Ichiyanagi, K., Pendall, E., Pu, J., and Yu, W.: Oxygen-18 concentrations in recent precipitation and ice cores on the Tibetan Plateau, J. Geophys. Res., 108(D9), 4293, doi:10.1029/2002JD002173, 2003.

Tian, L., Yao, T., Li, Z., MacClune, K., Wu, G., Xu, B., Li, Y., Lu, A., and Shen, Y.: Recent rapid warming trend revealed from the isotopic record in Muztagata ice core, eastern Pamirs, J. Geophys. Res., 111, D13103, doi: 10.1029/2005JD006249, 2006.

Tzedakis, P. C.: Seven ambiguities in the Mediterranean palaeoenvironmental narrative, Quat. Sci. Rev., 26, 2042–2066, doi:10.1016/j.quascirev.2007.03.014, 2007.

Yao, T., Thompson, L. G., Mosley-Thompson, E., Yang, Z., Zhang, X., and Lin, P.: Climatological significance of $\delta^{18}O$ in north Tibetan ice cores, J. Geophys. Res., 101(D23), 29531-29537, doi: 10.1029/96JD02683, 1996.

Zhang, Z., Hou, S., and Yi, S.: The first luminescence dating of Tibetan glacier basal sediment, The Cryosphere, 12, 1-6, doi: 10.5194/tc-12-1-2018, 2018.

Zhong, Z. P., Solonenko, N.E., Gazitúa, M. C., Kenny, D. V., Mosley-Thompson, E., Rich, V. I., Van Etten, J. L., Thompson, L. G., and Sullivan, M. B.: Clean low-biomass procedures and their application to ancient ice core microorganisms, Front. Microbiol., 9, 1-15, doi: 10.3389/fmicb.2018.01094, 2018.

---

## Author Comment (AC2) · 15 Apr 2019

Dear Prof. Lonnie Thompson,

Many thanks for your thoughtful referee comments. Below is a point-to-point response to your comments. The original comments are in black, and our response is marked in blue.

Referee Comments on the paper by Hou et al., Apparent discrepancy of Tibetan ice core  $\delta^{18}$ O records may be attributed to misinterpretation of chronology, for The Cryosphere Discuss., https://doi.org/10.5194/tc-2018-295.

First, it is certainly good to see the recent interest in our work on the Guliya ice core record that was conducted in the 1990s. The community has come a long way since that time when the greatest challenge that Tandong Yao and I faced when drilling in that part of the world was the question of whether or not it would be possible to drill an ice core at those elevations and then keep it frozen during its transit across the Gobi desert. We didn't know at the time how that work would set the stage for all of those who have come along since those early days.

Regarding the time scales on the early Guliya cores, they raised as many questions as they answered and therefore our team returned to Guliya in 2015 where we successfully recovered 5 ice cores, 4 of which were drilled to bedrock. A recently published paper highlights the geophysical work conducted in the field (Kutuzov *et al.*, 2018). A primary goal of the 2015 drilling campaign was to better constrain the time-scale on the Guliya ice cap by taking advantage of additional, newer analytical approaches and applying them to the freshly drilled ice cores. A number of these analyses are focused specifically on dating the ice and are now underway.

Kutuzov, S., L. G. Thompson, I. Lavrentiev, and L. Tian. 2018. Ice thickness measurements of Guliya ice cap, western Kunlun Mountains (Tibetan Plateau), China, *Journal of Glaciology*, 64(248) 977–989, doi: 10.1017/jog.2018.91.

**Response:**

We share the same experience and challenge of drilling ice cores at such high elevations. An additional challenge is to set up a reliable chronology for these mountain ice cores, especially for their bottom sections due to the rapid thinning of the ice layers and the dynamic nature of mountain glaciers. At present, tens of ice cores to the bedrock have been recovered from the Tibetan Plateau, but so far only three of them (i.e., Dunde, Guliya and Puruogangri) have provided a continuous time series beyond the last two millennia. Even for these three ice cores, there is much inconsistency among their  $\delta^{18}$ O records

(Fig. 3 of our TCD manuscript). Therefore, more Tibetan ice core  $\delta^{18}$ O records with reliable chronologies, including the Chongce and the new 2015 Guliya ice cores, are extremely necessary to reconcile the inconsistency among the Tibetan ice core  $\delta^{18}$ O records.

As an invited referee for the paper by Hou *et al.*, I have addressed a number of the specific issues raised in the manuscript but in short the paper lacks sufficient quantitative support for the authors' conclusions. I hope that the following points will help the authors improve their manuscript. Response:

Many thanks for the thoughtful comments below. We believe that our detailed responses to your questions/comments show that our conclusion is reasonable and based on solid evidence.

**Specific comments:**

Lines 50-55: "The Guliya record has been widely used as a benchmark for numerous studies since its publication (e.g., Fang et al., 1999; Rahaman et al., 2009; Sun et al., 2012; Hou et al., 2016; Li et al., 2017; Saini et al., 2017; Sanwal et al., 2019). Its stable isotopic record suggests a cooling mid-Holocene based on its decreasing  $\delta^{18}O$  values during that period. However, this cooling mid-Holocene is not found in other Tibetan ice core records available so far."

The first sentence will be addressed below. The third sentence is misleading. The mid-Holocene cooling is very noticeable in Tibetan climate records that are not from ice cores. For example, the regional vegetation and climate changes during the Holocene have been reconstructed from a high-resolution pollen record preserved in a peat sequence from the Altai Mountains of Xinjiang, China (Zhang *et al.*, 2018, *Quaternary Science Reviews*, 201, 111-123). These vegetation phases indicate that the regional climate changed from a cold and dry early Holocene to a warmer and wetter early-mid Holocene followed by a cold and dry mid-Holocene, which transitioned to a cool and wet late Holocene with warm and dry conditions characterizing the last millennium. Below is a figure comparing the data in Figure 6 of the Zhang *et al.* paper (left) with Figure 3 (right) from the Hou *et al.* paper. Note that the Guliya  $\delta^{18}$ O record (blue) is more similar to the mean annual temperature (Figure 6, panel f, red star) than the Chongce  $\delta^{18}$ O record. It is also important to note that the Guliya ice core was not used to help establish the chronology of the pollen record.

The figure is a composite of Figure 6 (Zhang et al., 2018) and Figure 3 (Hou et al., unpublished). The records above, along with other examples given below, dispute Lines 136-140 ("This warming trend during the mid-Holocene is similar to recent paleoclimatic reconstructions in other parts of the world (Samartin et al., 2017; Marsicek et al., 2018). By comparison, it seems that the  $\delta^{18}O$  profile of the Guliya ice core, especially for the period of 6-7 kaBP to  $\sim$ 3 kaBP, is at odds with this warming trend during the mid-Holocene."). Here the authors are picking records from regions thousands of miles away in much different climate regimes to confirm the Chongce  $\delta^{18}$ O record (and time scale). The Samartin et al. records are from the Mediterranean while the Marsicek et al. records are from Europe and North America. Hou et al. (Lines 35-40) state that "Marsicek et al. (2018) recently presented temperature reconstructions derived from sub-fossil pollen across North America and Europe. These records show a general long-term warming trend for the Holocene until ~2 kaBP (thousand years before present), and records with cooling trends are largely limited to North Atlantic, implying varied regional climate responses to global drivers"). There are several publications that link North Atlantic climate to the climates of Central Asia and China. Although most of them discuss the linkages between precipitation and westerlies influenced by North Atlantic atmospheric and oceanic processes, papers such as Feng and Hu (2008, Geophysical Research Letters 35 doi: 10.1029/2007GL032484) present an argument that North Atlantic SST anomalies strongly affect the TP surface temperature and heat sources, at least in the last century.

There are other records that call into question their conclusions regarding Holocene climate variability as inferred from the Chongce cores. For example, Zhang and Feng (*Earth-Science Reviews*, 2018, 185, 847-869) presented a compilation of pollen records from the Altai Mountains and surrounding regions that show a mid-Holocene cooling trend. Below see their Figure 37 (note panel d) from their synthesis of regional pollen records.

This is Figure 37 from Zhang and Feng, 2018 which was cited above.

Another example that does not support the conclusions drawn from the Chongce ice core is an alkenone-based 21 ka paleotemperature record from Lake Balikun (43.60-43.73°N, 92.74- 92.84°E, 1570 masl). As shown in the figure below (see panel d), this lake record shows that in this region the peak summer temperature occurred at 8 ka and was followed by general cooling throughout the Holocene.

---

## Author Comment (AC3) · 6 Jun 2019

Tian Lide (Orcid ID: 0000-0002-8866-7744)
Jiang Wei (Orcid ID: 0000-0002-6355-7637)
Lu Zheng-Tian (Orcid ID: 0000-0002-1887-0018)

**$^{81}$Kr dating of the Guliya ice cap, Tibetan Plateau**

**Lide Tian[1,2,3,4*], Florian Ritterbusch[5], Ji−Qiang Gu[5], Shui-Ming Hu[5], Wei Jiang[5], Zheng-Tian Lu[5*], Di Wang [1], Guo-Min Yang[5]**

[1]Institute of International Rivers and Eco-security, Yunnan University, Kunming 650500, China.

[2]CAS Center for Excellence in Tibetan Plateau Earth Sciences, Chinese Academy of Sciences, Beijing 100101, China.

[3]College of Resource and Environment, University of Chinese Academy of Sciences, Beijing, 100190, China

[4]Yunnan Key Laboratory of International Rivers and Transboundary Eco−security, Yunnan University, Kunming 650091, China

[5]Hefei National Laboratory for Physical Sciences at the Microscale, CAS Center for Excellence in Quantum Information and Quantum Physics, University of Science and Technology of China, Hefei 230026, China

Corresponding authors:

 Lide Tian (ldtian@ynu.edu.cn) , Zheng-Tian Lu (ztlu@ustc.edu.cn)

**Key Points**:

- Radiometric $^{81}$Kr dating of bottom ice from the Guliya ice cap

- $^{81}$Kr data yield upper age limits in the range of 15-74 ka

- $^{81}$Kr results difficult to reconcile with ages previously obtained for the Guliya ice core based on $^{36}$Cl and $\delta^{18}$O

This article has been accepted for publication and undergone full peer review but has not been through the copyediting, typesetting, pagination and proofreading process which may lead to differences between this version and the Version of Record. Please cite this article as doi: 10.1029/2019GL082464

© 2019 American Geophysical Union. All rights reserved.

**Abstract**

We present radiometric $^{81}$Kr dating results for ice samples collected at the outlets of the Guliya ice cap in the western Kunlun Mountains of the Tibetan Plateau. This first application of $^{81}$Kr dating on mid-latitude glacier ice was made possible by recent advances in Atom Trap Trace Analysis, particularly a reduction in the required sample size down to 1 μLSTP of krypton. Eight ice blocks were sampled from the bottom of the glacier at three different sites along the southern edges. The $^{81}$Kr data yield upper age limits in the range of 15-74 ka (90% confidence level). This is an order of magnitude lower than the ages exceeding 500 ka which the previous $^{36}$Cl data suggest for the bottom of the Guliya ice core. It is also significantly lower than the widely used chronology up to 110 ka established for the upper part of the core based on ice $\delta^{18}$O.

Keywords:$^{81}$K dating; Guliya ice cap; ice core chronology; Tibetan Plateau;

**Plain Language Summary**

The oldest ice that has ever been found outside of the polar regions is from the bottom of the Guliya ice cap in the western Kunlun mountains on the Tibetan Plateau. Previous dating results from the Guliya ice core, drilled in 1992, indicate that the bottom ice is more than 500 thousand years old. However, due to the lack of alternative dating methods in that time range, the age scale of the Guliya ice core has so far not been checked independently.

In this work we present dating results for the Guliya ice cap with a new dating method for ice based on the radioactive decay of the extremely rare krypton isotope $^{81}$Kr. Eight ice samples were retrieved at three different margin sites of the Guliya ice cap, where the old bottom ice is expected to resurface. The $^{81}$Kr measurements for these samples yield upper age limits in the range of 15-74 ka, which is significantly lower than the previous dating results for the ice core.

© 2019 American Geophysical Union. All rights reserved.

**1 Introduction**

Alpine ice cores in the mid- and low-latitude regions provide high-resolution records of past climate and environment. High rates of ice accumulation and melting are responsible for the relatively short history of ice core records on the Tibetan Plateau as compared to the polar regions. Longer ice cores and older ice are being sought on the Tibetan Plateau for the purpose of extending the climate history in this region. The Malan and Puruogangri ice cores in the central Tibetan Plateau [*Thompson et al.*, 2006; *Wang et al.*, 2003] and the Dasuopu ice core in the middle of the Himalayas [*Thompson et al.*, 2000; *Yao et al.*, 2002] provide records of the past several thousand years. Samples from the bottom of the Dunde ice core in the northeastern Tibetan Plateau were first interpreted to be glacial-stage ice [*Thompson et al.*, 1989], but later proved to be a Holocene deposit [*Thompson et al.*, 2005]. The longest (308.6 m) ice core and the oldest bedrock ice so far discovered on the Tibetan Plateau is from the Guliya ice cap in the western Kunlun Mountains [*Yao et al.* 1997, *Thompson et al.* 1997]. Developing a chronology for this Guliya ice core (GIC1992 hereafter), as for Tibetan ice cores in general, is challenging. Dating by layer counting is difficult for ice cores from the Tibetan Plateau because the monsoonal type precipitation pattern in this region generates weaker seasonal variation [*Hou et al.*, 2003]. For GIC1992 an age scale up to 110 ka was established down to 266 m depth by comparing the $\delta^{18}$O signal with the $CH_4$ record from GISP2 in Greenland. Moreover, the $^{36}$Cl data suggest that the bottom ice may be older than 500 ka. Since then, the GIC1992 record has been widely used as a reference for correlating regional climate signals [*e.g. Cheng et al.*, 2012; *Chevalier et al.*, 2011; *Cosford et al.*, 2008; *Hayashi et al.*, 2017; *Mahowald et al.*, 2011].

However, the established Guliya chronology is difficult to reconcile with several recent findings. *Cheng et al.* [2012] encountered inconsistencies between the $\delta^{18}$O record of GIC1992 and the Kesang stalagmite record. Their work suggests that the relationship between $\delta^{18}$O and $CH_4$ may be inversed, leading to a shortening of the GIC1992 age scale by a factor of two. Meanwhile, at the Chongce ice cap (~30 km away from the GIC1992 drilling site), luminescence dating provides an upper age limit of 42±4 ka for the basal sediment [*Zhang et al.*, 2018], which is an order of magnitude lower than what the $^{36}$Cl data suggests for the bottom ice of GIC1992. Moreover, $^{14}$C dating in combination with ice flow modeling for ice cores

© 2019 American Geophysical Union. All rights reserved.

from the Chongce ice cap indicates Holocene deposition [*Hou et al.*, 2018], which is consistent with all other Tibetan ice cores except GIC1992. Given the proximity between the Guliya and the Chongce ice cap, these results make it difficult to argue that the large difference in age scale between GIC1992 and the other Tibetan ice cores is due to different local climate conditions in the western Kunlun Mountains [*Thompson et al.,* 2005]. All the foregoing findings raise the need for examining the GIC1992 chronology with an independent dating method.

$^{81}$Kr is a cosmogenic radionuclide with a half-life of $229 \pm 11$ ka. The $^{81}$Kr concentration in the atmosphere (isotopic abundance $^{81}$Kr /Kr $\sim 1 \times 10^{-12}$) is spatially homogeneous with only small changes over the past 1.5 million years [*Buizert et al.*, 2014]. These properties as well as its chemical inertness make it a desirable tracer for groundwater and ice over the age range of 40 ka to 1.3 Ma [*Loosli and Oeschger*, 1969; *Lu et al.*, 2014]. Meanwhile, the anthropogenic $^{85}$Kr (half-life $10.76 \pm 0.02$ a), which is mainly produced by nuclear fuel reprocessing, can be used to identify any young ($< 60$ a) components or contamination of an old sample with modern air [*Winger et al.*, 2005]. Development of the analytical method of Atom Trap Trace Analysis (ATTA) has made radiokrypton dating available to the earth science community at large [*Jiang et al.*, 2012]. Due to the large required sample size (5-10 µL STP of krypton), so far $^{81}$Kr has been used mainly for dating groundwater while for glacier ice only a demonstration study was conducted on large blue ice samples (~ 350 kg) from Taylor Glacier, Antarctica [*Buizert et al.,* 2014]. Recently, the required sample size for $^{81}$Kr- and $^{85}$Kr-analysis has been reduced down to 1 µL STP of krypton, which can be extracted from about 10 kg of Antarctic ice (containing ~100 mL STP air per kg ice) or $20 - 40$ kg of Tibetan glacier ice (25 - 50 mL STP air/kg) [*Li et al.*, 2011]. This sample size is still too large to re-assess the historic GIC1992 directly, but is sufficient for $^{81}$Kr dating of samples from the margin sites of the Guliya ice cap, as presented in this work.

**2 Methods**

**2.1 Site description and ice sampling**

Guliya is a large ice cap in the western Kunlun Mountains on the Tibetan Plateau with a total area of about 376 km$^2$ [*Thompson et al.*, 1997; *Yao et al.*, 1997]. Its southern part is of nonsurge

© 2019 American Geophysical Union. All rights reserved.

type with stationary terminus positions [*Yasuda and Furuya*, 2015]. Remote sensing data show that the glaciers in this region have experienced less change in recent decades compared to other glaciated mountainous regions in western China [*Shangguan et al.*, 2017]. The Guliya ice cap even gained mass from 2000-2015 [*Kutuzov et al.*, 2018] primarily due to increasing precipitation in the westerly regime [*Yao et al.*, 2012]. Ice core drilling and ground penetrating radar show that the glacier thickness varies from about 50 m at the summit to a maximum thickness of 371 m at a location 1.5 km upstream of the GIC1992 drilling site (Fig. 1) [*Kutuzov et al.*, 2018; *Thompson et al.*, 1997]. The glacier flows from the summit at 6710 m altitude down to the margins at approximately 5500 m [*Thompson et al.,* 1997] with an average slope of < 3 -5° [*Kutuzov et al.*, 2018]. Limited field observation indicates increasing negative surface mass balance going from the equilibrium line altitude of around 6000 m to lower elevation sites [*Li et al.*, 2019]. The ablation of the ice cap is also characterized by cliff melting at the end of the glacier outlets so that the bottom ice layers become accessible over large sections of the glacier edge.

The criteria used in selecting sampling sites include less crevasse in the upper-stream ice flow and exposure of basal ice. The samples were collected at the bases of the vertical ice walls at three glacier outlets between 2015 and 2017. GLY1 is located downstream of the GIC1992 drilling site from 1992 [*Thompson et al.*, 1997]; GLY2 and GLY3 are located at the outlets of the glacier summit (Fig. 1). In total eight ice blocks were sampled, of which four were retrieved from a 6 m deep cave (hereafter cave samples) to avoid potential gas loss and contamination with modern air due to ice fractures [*Craig et al.,* 1990; *Buizert et al.*, 2014]. As the $^{85}$Kr results demonstrated that this practice is unnecessary at these sites, the later four ice samples were retrieved near the surface of the ice wall (hereafter surface samples).

At GLY1 (Fig. 1), a 6 meter deep horizontal cave was dug using a chainsaw and a pick along a clear bubble ice layer near the bottom of the ice wall. The underlying 2 m section contains dust layers with dark mud and pebbles and is thus not suitable for sampling with a chainsaw. Two vertically adjacent ice block samples (GLY1-1 on top of GLY1-2) were collected in the white bubble layer at the end of the cave. At GLY2, also a 6 m deep ice cave was dug approximately 3 m above the bottom dust layer, and two ice blocks were collected in the bubble ice layer.

© 2019 American Geophysical Union. All rights reserved.

Moreover, two near-surface glacier ice samples were retrieved along the ice cliff in order to compare dating results between cave and surface ice as well as to date the silty ice at the very bottom. At GLY3, silty ice blocks from the very bottom were collected (Fig. 2).

In addition to the samples for radiokrypton dating, ice samples for stable water isotope analysis were collected along the bottom of the ice wall surface. The surface layers were removed in the field since they may have been affected by melting. A glacier ice column with a total length of more than 5.3 m was sampled to the visible lowermost part of the glacier cliff at GLY2. A 1.27 m long ice column was sampled at GLY3 in the same layer where the ice blocks for $^{81}$Kr analysis had been sampled (Fig. 2). All the ice samples were kept in a freezer at -20℃ during the transport to the city of Lhasa, where they were stored in a cold room until degassing or cutting. The ice samples for $\delta^{18}$O analysis from GLY1 were lost due to technical failure of the cooling facility such that only the $\delta^{18}$O profiles from GLY2 and GLY3 are presented in the following.

**2.2 Air extraction from the ice samples**

For $^{81}$Kr and $^{85}$Kr analysis, the air trapped in the ice has to be extracted. Prior to extraction, the surface of the ice samples is cleaned to remove any layers or flaky debris that may contain modern air. The ice is then brought out of the cold room and placed in a stainless steel chamber which is thereafter sealed and evacuated for about 30 min by scroll pumps through a water trap (stainless steel bellow immersed in ethanol at -80 $^{o}$C). Since during pumping the chamber is constantly being flushed by the water vapor from the sublimating ice, the remaining atmospheric gas in the container is rendered negligible. After evacuation, the chamber is heated by a stove for 60 - 90 min (depending on the ice mass) until the ice has completely melted. The gas released from the ice passes through the water trap and is compressed into a sample cylinder. The air content of the ice sample is determined based on the final pressure in the sample cylinder (Table 1). Extraction efficiencies higher than 95% and contamination with modern air below 1% are typically achieved with this degassing method at a processing time of about 2-3 hours per sample. More details on the extraction system and procedure are provided in the supporting material.

© 2019 American Geophysical Union. All rights reserved.

**2.3 Krypton purification and $^{81}$Kr measurement**

The extracted gas from the ice samples was sent to the University of Science and Technology of China (USTC) for krypton purification and for ATTA analysis of both $^{81}$Kr and $^{85}$Kr. Krypton is separated from the extracted gas using a purification system based on titanium gettering and gas chromatography [*Tu et al.,* 2014], typically yielding krypton purities and recoveries both higher than 90%.

The $^{81}$Kr and $^{85}$Kr measurements are performed with the latest ATTA instrument at USTC, where individual $^{81}$Kr and $^{85}$Kr atoms are selectively laser-cooled and then detected in a magneto-optical trap. The stable and abundant $^{83}$Kr is also measured for normalization. The resulting $^{81}$Kr/$^{83}$Kr and $^{85}$Kr/$^{83}$Kr ratios for the sample are compared to the corresponding ratios of a reference krypton gas to derive the $^{81}$Kr abundance as a percentage of the atmospheric value (pMKr) and the $^{85}$Kr abundance given in the units of dpm/cc (decay per minutes per cc STP krypton), a convention originating from decay counting. More details on $^{81}$Kr and $^{85}$Kr analysis with ATTA can be found in *Jiang et al* [2012].

**2.4 Stable water isotopes**

The ice columns collected along the ice cliff at GLY2 and GLY3 were cut in the cold room into samples of 2 cm intervals. The melted ice samples were measured using Picarro L2140i liquid water analyzer in the Institute of International River and Eco-security, Yunnan University, with a precision of ±0.15‰ for $\delta^{18}$O referenced to VSMOW2.

**3 Results and discussion**
**3.1 $\delta^{18}$O results**

Figure 2 shows the oxygen isotope variation along the 5.3 m bottom ice at GLY2 and the 1.27 m profile at GLY3. It is difficult to match these short $\delta^{18}$O profiles from GLY2 and GLY3 with the $\delta^{18}$O record from GIC1992 [*Thompson et al.,* 1997]. However, the $\delta^{18}$O fluctuations along the profile provide hints whether the ice originates from the bottom or not. The accumulation layers of the glacier rapidly become thinner towards the bottom. The fast fluctuations of the $\delta^{18}$O signal is averaged out when the thickness of the layers become less than the 2 cm cutting

© 2019 American Geophysical Union. All rights reserved.

interval. The large fluctuations in the $\delta^{18}$O profile at GLY3 as well as in the upper part of GLY2 are comparable to those at the top of GIC1992 [*Thompson et al.*, 2018], suggesting that these samples are not derived from the bottom of the glacier. The samples were collected from the visible lowest part of the glacier cliff, which is not necessarily the lowest part of the ice as the bottom may be covered by debris from the glacier. This explanation is supported by the observation of a large amount of pebbles being deposited in front of the glacier cliff at GLY3. In contrast, the fluctuations in the $\delta^{18}$O profile at GLY2 exhibit reduced fluctuations towards the bottom of the glacier. This indicates that GLY2-4, collected at the bottom of the GLY2 profile, is likely close to the very bottom of the glacier ice. The reduced fluctuations in the lower 3.3 m of the $\delta^{18}$O profile at GLY2 may also result from mixing of ice of different ages due to complex flow leading to averaging of the $\delta^{18}$O values. The same mechanism may be responsible for the higher fluctuations in the $\delta^{18}$O profile at GLY3 and at the top of GLY2 (e.g. if layers with higher $\delta^{18}$O values are transported next to layers with lower $\delta^{18}$O values) although no stratigraphic disturbance has been observed at the three sampling sites.

It is difficult to match the $\delta^{18}$O records from this study to the one from GIC1992 because of the high ambiguity in matching the excursions and because the ice at GLY2 and GLY3 originates from a different accumulation zone than the ice at the GIC1992 drilling site. At the height of 3.3 m the $\delta^{18}$O record at GLY2 exhibits a shift in the mean from -17.5‰ to -15‰ and below that the standard deviation is reduced from 2‰ to 1.1‰ (figure 2). This behavior is similar for the $\delta^{18}$O signal of GIC1992 with the difference that the mean $\delta^{18}$O value at the bottom 40 m is higher than the bottom 3.3 m at GLY2 by about 2‰. This is likely due to the altitude difference of the accumulation zones of the ice at GLY2 and GIC1992.

**3.2 Air content**

The measured air contents in the ice samples are listed in Table 1. They vary from 32 mL STP/kg to 59 mL STP/kg, which is typical for Himalayan ice cores [*Hou et al.,* 2007; *Li et al.*, 2011] and significantly lower than the air content of Antarctic ice, typically ranging between 100 - 120 mL STP/kg [*Buizert et al.,* 2014*; Raynaud and Lebel*, 1979], or that of Greenland ice

© 2019 American Geophysical Union. All rights reserved.

at 80 - 100 mL/kg [*Raynaud et al.*, 1997]. This is due to the lower air pressure at high elevation (5500-6700 m) of the deposition site and the higher temperature compared to Antarctica [*Eicher et al.*, 2016; *Martinerie et al.*, 1992]. We deliberately collected the samples from ice layers with visibly high bubble content and avoided those with transparent ice which are likely layers of re-frozen meltwater.

**3.3 $^{85}$Kr and $^{81}$Kr results**

The measured $^{81}$Kr and $^{85}$Kr abundances for the eight glacier ice samples as well as two air samples of Lhasa are listed in Table 1. As described above, the $^{85}$Kr in the atmosphere has almost exclusively been produced anthropogenically in the past 60 years. Therefore, any sample older than that should have a vanishing $^{85}$Kr abundance. Five of the eight samples have $^{85}$Kr activity levels below 3% of the Lhasa air value (Table 1), whereas GLY2-1, GLY2-2 and GLY3-2 have $^{85}$Kr values corresponding to about 8%, 3% and 5%, respectively. Air leaks are thoroughly investigated on instruments used in the degassing, purification and ATTA measurement, leading to the conclusion that contamination of modern air during these processes is below 1%. It thus seems more likely that modern air had already entered the ice prior to sampling, e.g. by cracking/melting and refreezing, as has been observed in earlier works on glacier ice close to the surface of margin sites [*Craig et al.,* 1990; *Buizert et al*., 2014]. Since there is no obvious correlation between $^{85}$Kr and whether the sample is from the surface or from a cave, potential contamination processes at the very front of the glacier ice cliff do not seem to be responsible for that.

Since the measured $^{81}$Kr abundances are close to the modern value of 100 pMKr, contamination of modern air at these low concentrations does not affect the reported $^{81}$Kr abundances within the given precisions. For all samples they are consistent with modern atmospheric $^{81}$Kr abundance within 1σ, except for GLY3-1 which still lies within a 2σ error. We translate the measured relative $^{81}$Kr abundances into $^{81}$Kr-ages using the Feldman-Cousins method [*Feldman and Cousins,* 1998]. As the $^{81}$Kr abundances are close to modern, this method yields upper age limits (90% confidence level) for the individual samples that range between 15-74 ka.

© 2019 American Geophysical Union. All rights reserved.

**3.4 Implication for the Guliya ice core chronology**

The obtained results for $^{81}$Kr and $\delta^{18}$O of the Guliya margin samples allow for a discussion in the context of the results from GIC1992 [*Thompson et al.,* 1997] (see introduction). The $^{81}$Kr measurements do not show evidence for ice older than 74 ka at the bottom of the sampled margin sites of the Guliya ice cap. For the samples from GLY1, where the ice from GIC1992 is expected to outcrop [*Kutuzov et al.,* 2018], the upper limits for the $^{81}$Kr age do not exceed 52 ka. For GLY2-4, whose $\delta^{18}$O profile exhibits bottom ice characteristics, the $^{81}$Kr results provide an upper age limit of only 25 ka. The obtained upper age limits do not necessarily rule out the existence of older ice somewhere else in the Guliya ice cap. It is possible that the old ice at the bottom of GIC1992 is frozen to the bedrock and does not flow out to the margin sites. However, radar measurements indicate that the ice at the bottom of GIC1992 does flow and is not trapped at the bedrock [*Kutuzov et al.*, 2018]. A further explanation is that the stratigraphy of the glacier ice is folded when travelling from the GIC1992 drilling site to the margin, such that the old ice may not be at the bottom. No evidence for folding was observed at the glacier terminals, which exhibit clear horizontal layer structures, but folding on intermediate distance scales may have occurred. Yet another possibility is that the bottom 100 m of GIC1992, which are supposedly older than 50 ka, are rapidly thinning towards the outlet of the glacier, and therefore may be contained in a much smaller vertical extent at the very bottom of the glacier cliff. Since the samples at GLY1 were taken in about 2 m height above bedrock, they may not reach into this old bottom section. However, measurements of the mass balance and the glacier surface velocity [*Thompson et al.*, 1995; *Li et al.*, 2019; *Chadwell*, 2017] indicate that a large fraction of the upper glacier layers is lost when flowing from the equilibrium line altitude to the edge of the glacier at GLY1 where the remaining glacier cliff is about 10 m in height. Therefore, it does not seem likely that the bottom 100 m at the GIC1992 drilling site are thinning to below our sampling height about 2 m above bedrock at GLY1.

© 2019 American Geophysical Union. All rights reserved.

**4 Conclusions and Outlook**

Radiometric $^{81}$Kr dating has been used to determine the age of bottom ice samples at the Guliya ice cap. Eight ice blocks, each weighing 28-69 kg, were collected at three different outlets of the glacier, and analyzed for $^{81}$Kr using the Atom Trap Trace Analysis method. The $^{81}$Kr results yield upper limits in the range of 15-74 ka, which is an order of magnitude lower than previously suggested by $^{36}$Cl dating of the Guliya ice core and also significantly lower than the Guliya chronology reaching up to 110 ka based on $\delta^{18}$O measurements. After results from the Kesang stalagmite cave (~ 860 km distance to the Guliya ice cap) and the Chongce ice cap (~ 30 km distance), the $^{81}$Kr data in this work (obtained directly from bottom samples of the Guliya ice cap) represent yet another result that calls for further dating measurements to check the established Guliya chronology. Measurements of $^{14}$C, $^{36}$Cl, $^{10}$Be, $\delta^{18}O_{atm}$ and argon isotope ratios are planned for a new Guliya ice core that has been drilled in 2015 close to the location of the 1992 Guliya core drilling site [*Thompson et al.,* 2018]. Meanwhile, at the USTC laboratory, work is in progress to further reduce the sample size required for $^{81}$Kr analysis so that bottom samples from the Guliya ice core can be measured directly.

**Acknowledgments**

We thank the two anonymous reviewers for their valuable comments and suggestions. This work is funded by National Natural Science Foundation of China (41530748), the National Key Research and Development Program of China (2016YFA0302200) and the Chinese Academy of Sciences (XDB21010200). We thank Lili Shao and Cheng Wang from the Institute of Tibetan Plateau Research for their assistance in the ice degassing and Lei Zhao from USTC for purifying the krypton samples. Stable water isotopes data are available in the supporting information.

© 2019 American Geophysical Union. All rights reserved.

**References**

Buizert, C., et al. (2014), Radiometric [81]Kr dating identifies 120,000-year-old ice at Taylor Glacier, Antarctica, Proceedings of the National Academy of Sciences of the United States of America, 111(19), 6876-6881, doi: 10.1073/pnas.1320329111.

Chadwell, C. (2017), Reliability analysis for design of stake networks to measure glacier surface velocity, Journal of Glaciology, 45(149), 154-164, doi: 10.1017/s0022143000003130.

Cheng, H., P. Z. Zhang, C. Spötl, R. L. Edwards, Y. J. Cai, D. Z. Zhang, W. C. Sang, M. Tan, and Z. S. An (2012), The climatic cyclicity in semiarid-arid central Asia over the past 500,000 years, Geophysical Research Letters, 39(1), doi: 10.1029/2011gl050202.

Chevalier, M.-L., G. Hilley, P. Tapponnier, J. Van Der Woerd, J. Liu-Zeng, R. C. Finkel, F. J. Ryerson, H. Li, and X. Liu (2011), Constraints on the late Quaternary glaciations in Tibet from cosmogenic exposure ages of moraine surfaces, Quaternary Science Reviews, 30(5), 528-554, doi: https://doi.org/10.1016/j.quascirev.2010.11.005.

Cosford, J., H. Qing, D. Yuan, M. Zhang, C. Holmden, W. Patterson, and C. Hai (2008), Millennial-scale variability in the Asian monsoon: Evidence from oxygen isotope records from stalagmites in southeastern China, Palaeogeography, Palaeoclimatology, Palaeoecology, 266(1), 3-12, doi: https://doi.org/10.1016/j.palaeo.2008.03.029.

Craig, H., T. E. Cerling , R. D. Willis, W. A. Davis, C. Joyner, N. Thonnard (1990), Krypton-81 in Antarctic ice: first measurement of a Krypton age on ancient ice. EOS 71:1825.

Eicher, O., M. Baumgartner, A. Schilt, J. Schmitt, J. Schwander, T. F. Stocker, and H. Fischer (2016), Climatic and insolation control on the high-resolution total air content in the NGRIP ice core, Climate of the Past, 12(10), 1979-1993, doi: 10.5194/cp-12-1979-2016.

© 2019 American Geophysical Union. All rights reserved.

Feldman, G.J. and R.D. Cousins (1998), Unified approach to the classical statistical analysis of small signals, Physical Review D, 57, 3873–3889, doi: 10.1103/PhysRevD.57.3873

Hayashi, T., et al. (2017), Ecological variations in diatom assemblages in the Paleo-Kathmandu Lake linked with global and Indian monsoon climate changes for the last 600,000 years, Quaternary Research, 72(3), 377-387, doi: 10.1016/j.yqres.2009.07.003.

Hou, S., D. Qin, J. Jouzel, V. Masson-Delmotte, U. Von Grafenstein, A. Landais, N. Caillon, and J. Chappellaz (2004), Age of Himalayan bottom ice cores, Journal of Glaciology, 50(170), 467-468, doi: 10.3189/172756504781829981.

Hou, S., J. Chappellaz, J. Jouzel, P. C. Chu, V. Masson-Delmotte, D. Qin, D. Raynaud, P. A. Mayewski, V. Y. Lipenkov, and S. Kang (2007), Summer temperature trend over the past two millennia using air content in Himalayan ice, Climate of the Past, 3(1), 89-95, doi: 10.5194/cp-3-89-2007

Hou, S., T. M. Jenk, W. Zhang, C. Wang, S. Wu, Y. Wang, H. Pang, and M. Schwikowski (2018), Age ranges of the Tibetan ice cores with emphasis on the Chongce ice cores, western Kunlun Mountains, The Cryosphere, 12(7), 2341-2348, doi: 10.5194/tc-12-2341-2018.

Jiang, W., et al. (2012), An atom counter for measuring $^{81}$Kr and $^{85}$Kr in environmental samples, Geochimica et Cosmochimica Acta, 91, 1-6, doi: 10.1016/j.gca.2012.05.019.

Kutuzov, S., L. G. Thompson, I. Lavrentiev, and L. Tian (2018), Ice thickness measurements of Guliya ice cap, western Kunlun Mountains (Tibetan Plateau), China, Journal of Glaciology, 1-13, doi: 10.1017/jog.2018.91.

Li, J., B. Xu, and J. Chappellaz (2011), Variations of air content in Dasuopu ice core from AD 1570–1927 and implications form climate change, Quaternary International, 236(1-2), 91-95, doi: 10.1016/j.quaint.2010.05.026.

Li, S., T. Yao, W. Yu, W. Yang, and M. Zhu (2019), Energy and mass balance characteristics of the Guliya ice cap in the West Kunlun Mountains, Tibetan Plateau, Cold Regions Science and Technology, 159, 71-85, doi: 10.1016/j.coldregions.2018.12.001.

Loosli, H. H., and H. Oeschger (1969), $^{37}$Ar and $^{81}$Kr in the atmosphere, Earth and Planetary

© 2019 American Geophysical Union. All rights reserved.

Science Letters, 7(1), 67-71, doi: 10.1016/0012-821X(69)90014-4.

Lu, Z. T., et al. (2014), Tracer applications of noble gas radionuclides in the geosciences, Earth-Science Reviews, 138, 196-214, doi: 10.1016/j.earscirev.2013.09.002.

Mahowald, N., S. Albani, S. Engelstaedter, G. Winckler, and M. Goman (2011) Model insight into glacial–interglacial paleodust records, Quaternary Science Reviews, 30(7-8), 832-854, doi: 10.1016/j.quascirev.2010.09.007.

Martinerie, P., D. Raynaud, D. M. Etheridge, J.-M. Barnola, and D. Mazaudier (1992), Physical and climatic parameters which influence the air content in polar ice, Earth and Planetary Science Letters, 112(1), 1-13, doi: 10.1016/0012-821X(92)90002-D.

Raynaud, D., and B. Lebel (1979), Total gas content and surface elevation of polar ice sheets, Nature, 281(5729), 289-291, doi: 10.1038/281289a0.

Raynaud, D., J. Chappellaz, C. Ritz, and P. Martinerie (1997), Air content along the Greenland Ice Core Project core: A record of surface climatic parameters and elevation in central Greenland, Journal of Geophysical Research: Oceans, 102(C12), 26607-26613, doi:10.1029/97JC01908.

Shangguan, D., S. Liu, Y. Ding, J. Li, Y. Zhang, L. Ding, X. Wang, C. Xie, and G. Li (2017), Glacier changes in the west Kunlun Shan from 1970 to 2001 derived from Landsat TM/ETM+ and Chinese glacier inventory data, Annals of Glaciology, 46(1), 204-208, doi: 10.3189/172756407782871693.

Thompson, L. G., T. Yao, E. Mosley-Thompson, M. E. Davis, K. A. Henderson, and P. Lin (2000), A high-resolution millennial record of the south Asian monsoon from Himalayan ice cores, Science, 289(5486), 1916-1920, doi: 10.1126/science.289.5486.1916.

Thompson, L. G., M. E. Davis, E. Mosley-Thompson, P. N. Lin, K. A. Henderson, and T. A. Mashiotta (2005), Tropical ice core records: evidence for asynchronous glaciation on Milankovitch timescales, Journal of Quaternary Science, 20(7-8), 723-733, doi: 10.1002/jqs.972.

Thompson, L. G., E. Mosley-Thompson, M. E. Davis, T. A. Mashiotta, K. A. Henderson, P. N. Lin, and T. Yao (2006), Ice core evidence for asynchronous glaciation on the

© 2019 American Geophysical Union. All rights reserved.

Tibetan Plateau, Quaternary International, 154(0), 3-10, doi: 10.1016/j.quaint.2006.02.001.

Thompson, L. G., E. Mosley-Thompson, M. E. Davis, J. F. Bolzan, J. Dai, L. Klein, T. Yao, X. Wu, Z. Xie, and N. Gundestrup (1989), Holocene--late Pleistocene climatic ice core records from Qinghai-Tibetan plateau, Science, 246(4929), 474-477, doi: 10.1126/science.246.4929.474.

Thompson, L. G., T. Yao, M. E. Davis, K. A. Henderson, E. Mosley-Thompson, P. N. Lin, J. Beer, H. A. Synal, J. ColeDai, and J. F. Bolzan (1997), Tropical climate instability: The last glacial cycle from a Qinghai-Tibetan ice core, Science, 276(5320), 1821-1825, doi: 10.1126/science.276.5320.1821.

Thompson, L. G., et al. (2018), Ice core records of climate variability on the Third Pole with emphasis on the Guliya ice cap, western Kunlun Mountains, Quaternary Science Reviews, 188, 1-14, doi: 10.1016/j.quascirev.2018.03.003.

Tu, L.-Y., G.-M. Yang, C.-F. Cheng, G.-L. Liu, X.-Y. Zhang, and S.-M. Hu (2014), Analysis of Krypton-85 and Krypton-81 in a Few Liters of Air, Analytical Chemistry, 86(8), 4002-4007, doi: 10.1021/ac500415a.

Wang, N., L. G. Thompson, M. E. Davis, E. Mosley-Thompson, T. Yao, and J. Pu (2003), Influence of variations in NAO and SO on air temperature over the northern Tibetan Plateau as recorded by $\delta^{18}O$ in the Malan ice core, Geophysical Research Letters, 30(22), CLM 1-5, doi: 10.1029/2003gl018188.

Winger, K., J. Feichter, M. B. Kalinowski, H. Sartorius, and C. Schlosser (2005), A new compilation of the atmospheric Krypton-85 inventories from 1945 to 2000 and its evaluation in a global transport model, Journal of Environmental Radioactivity, 80(2), 183-215, doi: 10.1016/j.jenvrad.2004.09.005.

Yao, T., L. G. Thompson, and Y. Shi (1997), Climatic variation since the Last Interglacial recorded in the Guliya ice core, Science in China (D), 40(6), 662-668, doi: 10.1007/BF02877697.

© 2019 American Geophysical Union. All rights reserved.

Yao, T., et al. (2012), Different glacier status with atmospheric circulations in Tibetan Plateau and surroundings, Nature Climate Change, 2(9), 663-667, doi: 10.1038/Nclimate1580.

Yao, T., K. Duan, B. Xu, N. Wang, J. Pu, S. Kang, X. Qin, and L. G. Thompson (2002), Temperature and methane changes over the past 1000 years recorded in Dasuopu glacier (central Himalaya) ice core, Annals of Glaciology, Vol 35, 35, 379-383.

Yasuda, T., and M. Furuya (2015), Dynamics of surge-type glaciers in West Kunlun Shan, Northwestern Tibet, Journal of Geophysical Research: Earth Surface, 120(11), 2393-2405, doi: 10.1002/2015jf003511.

Zhang, Z., S. Hou, and S. Yi (2018), The first luminescence dating of Tibetan glacier basal sediment, The Cryosphere, 12(1), 163-168, doi: 10.5194/tc-12-163-2018.

© 2019 American Geophysical Union. All rights reserved.

**Table 1**. Compilation of the [81]Kr and [85]Kr results. The [81]Kr abundance is reported in units of pMKr (percent Modern Krypton). The atmospheric level is 100 pMKr. The [85]Kr abundance is reported in the units of dpm/cc (decays per minute per cc STP of krypton). The errors are 1σ standard deviations whereas upper limits are reported for a 90% confidence level.

| Sample | Note | Weight kg | Air content mL STP/kg | Krypton μL STP | [85]Kr dpm /cc | [81]Kr pMKr | [81]Kr age ka |
|---|---|---|---|---|---|---|---|
| GLY1-1 | Cave | 34 | 52 | 1.4 | < 1.5 | 97 ± 7 | < 52 |
| GLY1-2 | Cave | 28 | 46 | 1.3 | 1.6 ± 0.2 | 106 ± 6 | < 15 |
| GLY2-1 | Cave | 53 | 37 | 1.1 | 6.1 ±1.8 | 93 ±7 | < 74 |
| GLY2-2 | Cave | 69 | 41 | 2.5 | 2.5 ±0.2 | 97 ± 5 | < 39 |
| GLY2-3 | Surface | 52 | 45 | 1.7 | 0.7 ± 0.2 | 97 ± 5 | < 39 |
| GLY2-4 | Surface | 30 | 32 | 0.7 | < 0.4 | 104 ± 7 | < 25 |
| GLY3-1 | Surface | 43 | 29 | 1.8 | 1.0 ± 0.2 | 93 ± 5 | < 58 |
| GLY3-2 | Surface | 36 | 50 | 1.4 | 4.1 ± 0.4 | 98 ± 6 | < 45 |
| Lhasa-Air1 | May 2017 | - | - | 0.9 | 75 ± 2 | - | - |
| Lhasa-Air2 | Oct 2017 | - | - | 0.9 | 76 ± 3 | - | - |

© 2019 American Geophysical Union. All rights reserved.

[Figure]

**Figure 1**. (a) Location of the Guliya ice cap on the Tibetan Plateau ; (b) photograph showing the glacier cliff (~20 m tall) at sampling site GLY3; (c) Sampling sites GLY1, GLY2 and GLY3 for bottom ice of the Guliya ice cap during 2015-2017. The red dot marks the summit (6710 m a.s.l.) and the red star the location of the Guliya ice core (GIC1992) drilling site (6200 m a.s.l.) from 1992 [*Thompson et al.,* 1997].

© 2019 American Geophysical Union. All rights reserved.

[Figure]

**Figure 2**. Vertical $\delta^{18}O$ profiles along a 5.3 m column at GLY2 and a 1.27 m column at GLY3. The boxes show the positions of the $^{81}$Kr-dated glacier ice samples along the vertical profiles. The zero in height corresponds to the visible bottom of the glacier cliff, but is not necessarily the bedrock as debris may cover the lowermost part of the glacier. The size and the vertical position of the samples are roughly to scale. For GLY2, the $\delta^{18}O$ data of the lower 3.3 m have an average of -15.0‰ and a standard deviation (std) of 1.1‰ whereas in the upper 2 m the average is -17.5‰ (std=2.0‰). For GLY3, the average is -16.6‰ (std=2.4‰).

© 2019 American Geophysical Union. All rights reserved.

---

## Author Response (AR3)

Dear Dr. Martin,

Many thanks for the thoughtful comments from you and the anonymous Referee #1. Below I have provided a point-to-point response to these comments. The comments are in black, and the response is in blue. I have revised the manuscript accordingly.

Sincerely yours,

Hou Shugui

Comments to the Author:

Dear authors,

Thanks again for your submission to TC/TCD and for the effort you have put so far in the revision of the manuscript. I agree with the reviewers that the manuscript has improved considerably.

The paper touches a few controversial points: ice-core data accessibility,  $\delta^{18}$ O as a proxy for temperature, Holocene climate variability and synchronicity of glaciation of the Tibetan Plateau. The discussion with the reviewers has been really interesting and I have found it very instructive.

It is clear to me that the paper is not going to end the controversy in these topics. It does not need to. My view is that it is worth to put forward hypothesis, in this case to reconcile Tibetan Plateau ice cores interpretations, as far as there is a balanced discussion that includes competing hypothesis.

I am asking the authors to check again the reviewers comments and make sure that the manuscript incorporates a balanced discussion. I don't mean that the authors have to strengthen their hypothesis and include more references. I mean that the authors present with more clarity the different views and what is the evidence underlying them. Reviewer #1 highlights the interpretation of  $\delta^{18}$ O as proxy for temperature. Reviewer #2 challenges the derived Holocene climate variability and synchronicity of glaciation derived from the Chongce glacier.

**Response:**

I fully agree that this paper is not going to end any of the controversies, but rather it tries to draw an attention to the apparent discrepancy of Tibetan ice core  $\delta^{18}$ O records and provide a possible explanation. In the revision, we have kept the discussion as neutral as possible by providing perspectives from different sides.

Regarding Reviewer #1's comments on the interpretation of  $\delta^{18}$ O as a proxy for temperature, we added the following text:

Lines 139-148:

There is still uncertainty in the interpretation of the  $\delta^{18}$ O data of Tibetan ice cores solely as a temperature proxy across the entire Holocene period. In addition to local temperature, the precipitation  $\delta^{18}$ O could be affected by other factors in longer timescales such as changes in the regional circulation patterns, moisture sources and shifts in seasonal distribution of precipitation (Cheng et al., 2016, Ren et al., 2017). Therefore, more studies are needed to further examine the validity of using ice core  $\delta^{18}$ O as a temperature proxy on the TP. Nevertheless, simulations by the isotopic general circulation model (LMDZiso) indicate that a strong positive correlation exists between the local temperature and precipitation isotope, and it has persisted during the Holocene (Risi et al., 2010).

- Cheng, H., Spötl, C., Breitenbach, S. F. M., Sinha, A., Wassenburg, J. A., Jochum, K. P., Scholz, D., Li, X., Yi, L., Peng, Y., Lv, Y., Zhang, P., Votintseva, A., Loginov, V., Ning, Y., Kathayat, G., and Edwards, R. L.: Climate variations of central Asia on orbital to millennial timescales, Sci. Rep., 6, 36975, https://doi.org/10.1038/srep36975, 2016.
- Ren, W., Yao, T., Xie, S., and He, Y.: Controls on the stable isotopes in precipitation and surface waters across the southeastern Tibetan Plateau, J. Hydrol., 545, 276-287, https://doi.org/10.1016/j.jhydrol.2016.12.034, 2017.
- Risi, C., Bony, S., Vimeux, F., and Jouzel, J.: Water stable isotopes in the LMDZ4
  General Circulation Model: Model evaluation for present day and past climates
  and applications to climatic interpretation of tropical isotopic records, J.
  Geophys. Res., 115, D12118, https://doi.org/10.1029/2009jd013255, 2010.

In response to Reviewer #2's comments on the Holocene climate variability and synchronicity of glaciation derived from the Chongce glacier, we added the following text:

Lines 185-189:

We are aware of studies suggesting a mid-Holocene cooling trend on the TP and surrounding regions, as argued by Thompson (2019 and references therein). Meanwhile, other recent studies show a warming trend (e.g., Rao et al., 2019), similar to our results. Therefore, more research is needed to reach a more definitive conclusion on the mid-Holocene climate variations.

**Lines 224-228:**

Although the synchronicity of glaciation on the TP is beyond the scope of the current work, our new understanding of the Guliya ice core chronology would cast doubt on using the Guliya record based on its original chronology as supporting evidence for asynchronous glaciation on the TP on Milankovitch timescales (Thompson et al., 2005).

- Rao, Z., Wu, D., Shi, F., Guo, H., Cao, J., and Chen, F.: Reconciling the 'westerlies' and 'monsoon' models: A new hypothesis for the Holocene moisture evolution of the Xinjiang region, NW China, Earth-Sci. Rev., 191, 263-272, https://doi.org/10.1016/j.earscirev.2019.03.002, 2019.
- Thompson, L. G., Davis, M., Mosley-Thompson, E., Lin, P., Henderson, K., and Mashiotta, T.: Tropical ice core records: evidence for asynchronous glaciation on Milankovitch timescales, J. Quat. Sci., 20, 723-733, https://doi.org/10.1002/jqs.972, 2005.
- Thompson, L.: Interactive comment on "Apparent discrepancy of Tibetan ice core δ18O records may be attributed to misinterpretation of chronology" by Shugui Hou et al., The Cryosphere Discuss., https://doi.org/10.5194/tc-2018-295-RC2, 2019.

In addition, the discussion of the dating of the bottom samples at the Guliya ice cap will benefit from a reference to the recently accepted paper from Tian Lide et al in GRL. Response:

**The GRL paper of Tian Lide et al. was cited in the revision.**

**Anonymous Referee #1**

The revision has improved the manuscript considerably. I only have one minor suggestion:

I am not quite convinced about the arguments that the  $\delta^{18}$ O data of Tibetan ice cores can be solely interpreted as a temperature proxy across the entire Holocene period. It would be sound for authors to discuss an alternative interpretation: the precipitation  $\delta^{18}$ O change (e.g., the precession rhythm). This is observed across the vast westerlies and monsoon domains, including the upstream moisture of Tibetan ice cores, which should play an important role independent of the local temperature at the ice core sites. Closely related to the  $\delta^{18}$ O interpretation is another issue: the largest abrupt  $\delta^{18}$ O change (~8‰) observed in the Guliya record, or equivalently, the multi-decadal-scale mean temperature change of ~10°C or larger (?), around the relative distance of ~0.58 (Fig. 5), would be interesting, although difficult to understand, because this would suggest a number of huge temperature oscillations (on multi-decadal timescale?) reoccurred during the late Holocene.

**Response:**

We include the following text in the revision:

Lines 139-148:

There is still uncertainty in the interpretation of the  $\delta^{18}$ O data of Tibetan ice cores solely as a temperature proxy across the entire Holocene period. In addition to local temperature, the precipitation  $\delta^{18}$ O could be affected by other factors in longer timescales such as changes in the regional circulation patterns, moisture sources and shifts in seasonal distribution of precipitation (Cheng et al., 2016, Ren et al., 2017). Therefore, more studies are needed to further examine the validity of using ice core  $\delta^{18}$ O as a temperature proxy on the TP. Nevertheless, simulations by the isotopic general circulation model (LMDZiso) indicate that a strong positive correlation exists between the local temperature and precipitation isotope, and it has persisted during the Holocene (Risi et al., 2010).

As we discussed in our response to the previous comments, the large amplitude of  $\delta^{18}$ O variations could be attributed to the following two factors: (1) the elevation dependency

of temperature change observed in many studies for the TP, i.e. high altitude regions experience larger temperature changes than low elevation regions; (2) changes in water vapor sources associated with northward and southward shifts of the westerly circulation on longer timescale (e.g., from multi-millennial to orbital timescales). Beyond these factors, we agree that it is difficult to further understand the large amplitudes of  $\delta^{18}$ O variations, but a detailed discussion on this topic may be beyond the scope of the current manuscript. We revised the manuscript accordingly as follows:

**Lines 149-160:**

[revised manuscript text omitted]